# [Hexaamminecobalt(III)] Dichloride Permanganate—Structural Features and Heat-Induced Transformations into (Co$^{II}$,Mn$^{II}$)(Co$^{III}$,Mn$^{III}$)$_2$O$_4$ Spinels

Laura Bereczki [1,2], Vladimir M. Petruševski [3], Fernanda Paiva Franguelli [1,4], Kende Attila Béres [1,5], Attila Farkas [6], Berta Barta Holló [7], Zsuzsanna Czégény [1], Imre Miklós Szilágyi [4] and László Kótai [1,8,*]

1. Institute of Materials and Environmental Chemistry, Research Centre for Natural Sciences, Magyar Tudósok krt. 2., H-1117 Budapest, Hungary
2. Centre for Structural Science, Research Centre for Natural Sciences, Magyar Tudósok krt. 2., H-1117 Budapest, Hungary
3. Faculty of Natural Sciences and Mathematics, Ss. Cyril and Methodius University, MK-1000 Skopje, North Macedonia
4. Department of Inorganic and Analytical Chemistry, Budapest University of Technology and Economics, Műegyetem rakpart 3, H-1111 Budapest, Hungary
5. György Hevesy PhD School of Chemistry, Institute of Chemistry, ELTE Eötvös Loránd University, Pázmány Péter s. 1/A, H-1117 Budapest, Hungary
6. Department of Organic Chemistry and Technology, Budapest University of Technology and Economics, Budafoki út 8, H-1111 Budapest, Hungary
7. Department of Chemistry, Biochemistry and Environmental Protection, Faculty of Sciences, University of Novi Sad, Trg Dositeja Obradovića 3, 21000 Novi Sad, Serbia
8. Deuton-X Ltd., Selmeci u. 89, H-2030 Érd, Hungary
* Correspondence: kotai.laszlo@ttk.hu

**Abstract:** We synthesized and characterized (IR, Raman, UV, SXRD) hexaamminecobalt(III) dichloride permanganate, [Co(NH$_3$)$_6$]Cl$_2$(MnO$_4$) (compound **1**) as the precursor of Co–Mn–spinel composites with atomic ratios of Co:Mn = 1:1 and 1:3. The 3D−hydrogen bond network includes N–H···O–Mn and N–H···Cl interactions responsible for solid-phase redox reactions between the permanganate anions and ammonia ligands. The temperature-limited thermal decomposition of compound **1** under the temperature of boiling toluene (110 °C) resulted in the formation of (NH$_4$)$_4$Co$_2$Mn$_6$O$_{12}$, which contains a todorokite-like manganese oxide network (Mn$^{II}$$_4$Mn$^{III}$$_2$O$_{12}$$^{10−}$). The heat treatment products of compounds **1** and [Co(NH$_3$)$_5$Cl](MnO$_4$)$_2$ (**2**) synthesized previously at 500 °C were a cubic and a tetragonal spinel with Co$_{1.5}$Mn$_{1.5}$O$_4$ and CoMn$_2$O$_4$ composition, respectively. The heating of the decomposition product of compounds **1** and **2** that formed under refluxing toluene (a mixture with an atomic ratio of Co:Mn = 1:1 and 1:2) and after aqueous leaching ((NH$_4$)$_4$Co$_2$Mn$_6$O$_{12}$, 1:3 Co:Mn atomic ratio in both cases) at 500 °C resulted in tetragonal Co$_{0.75}$Mn$_{2.25}$O$_4$ spinels. The Co$_{1.5}$Mn$_{1.5}$O$_4$ prepared from compound **1** at 500 °C during the solid-phase decomposition catalyzes the degradation of Congo red with UV light. The decomposition rate of the dye was found to be nine times faster than in the presence of the tetragonal CoMn$_2$O$_4$ spinel prepared in the solid-phase decomposition of compound **2**. The todorokite-like intermediate prepared from compound **1** under N$_2$ at 115 °C resulted in a 54 times faster degradation of Congo red, which is a great deal faster than the same todorokite-like phase that formed from compound **2** under N$_2$.

**Keywords:** permanganate; ammine; solid-phase quasi-intramolecular redox reaction; todorokite; spinel; photochemical degradation; Congo red

## 1. Introduction

The preparation and thermal decomposition of transition metal complexes with reducing ligands and oxygen−containing anions are intensively studied areas of coordination

chemistry [1–22], especially due to the quasi-intramolecular redox reactions observed between their reducing ligands and oxidizing anions, which result in various simple and mixed nanosized metal oxides. If the oxidizing anion is permanganate, and the ligand is ammonia, the controlled−temperature thermal decomposition generally resulted in a $MMn_2O_4$ spinel and other bimetallic oxides [23–29]. The presence of halogens, depending on the chemical form and character in the complexes (ligand or anion) can drastically change the nature of the products. For example, diamminesilver perchlorate resulted in AgCl as the end−product [30], whereas chloride ligands in [pentaammine(chlorido)cobalt(III)] dipermanganate resulted in the expected phase−pure $CoMn_2O_4$ spinel [9] containing chloride ions and cobalt(III) in the redox interactions. The cobalt manganese oxide spinel compounds have enormous importance in catalysis; e.g., Mansouri et al. prepared a cobalt manganese oxide spinel (given as $CoMn_2O_4$) by the thermal decomposition of the $[Co(NH_3)_4CO_3]MnO_4$ complex [28,29] with excellent activity in the Fischer–Tropsch fuel synthesis. However, $[Co(NH_3)_4CO_3]MnO_4$ contains Co and Mn in a 1:1 atomic ratio. Thus, the decomposition product is probably a mixed $Co_{.1.5}Mn_{1.5}O_4$ spinel or a mixture of $CoMn_2O_4$ and $Co_3O_4$. Klobb prepared a compound, $[Co(NH_3)_6]Cl_2(MnO_4)$ (compound **1**) with a Co:Mn ratio of 1:1 [31]. Compound **1** is expected to transform Co–Mn–oxides with a Co:Mn = 1:1 ratio, and it allows the comparison of the catalytic properties of the $Co_{1.5}Mn_{1.5}O_4$ spinel phases prepared from compound **1** and $[Co(NH_3)_4CO_3]MnO_4$. Compound **1** has not been characterized in detail; therefore in the present paper, we discussed the structural and spectroscopic features and the influence of outer sphere chloride anions on its thermal decomposition, including the comparison of the effect of chloride position (outer or inner in compounds **1** and **2**, respectively) on the thermal and redox processes.

## 2. Results and Discussion

### 2.1. Synthesis and Properties of Compound **1**

[Hexaamminecobalt(III)] dichloride permanganate (compound **1**) has been isolated first by Klobb [31] as a by-product in the synthesis reaction of $[Co(NH_3)_6](MnO_4)_3$ (compound **3**) from $[Co(NH_3)_6]Cl_3$ (compound **4**) and $KMnO_4$ in water at 50 °C. Klobb isolated pure compound **1** in the reaction of compound **3** and a huge excess of $[Co(NH_3)_6]Cl_3$ in water at 50 °C. The previously isolated $Cl^-$, $MnO_4^-$ and $ClO_4^-$ containing compounds and their abbreviations are listed in Table 1.

$$[Co(NH_3)_6](MnO_4)_3 + 2[Co(NH_3)_6]Cl_3 = 3[Co(NH_3)_6]Cl_2MnO_4$$

**Table 1.** Labels of compounds.

| Compound | Label |
|---|---|
| $[Co(NH_3)_6]Cl_2(MnO_4)$ | **1** |
| $[Co(NH_3)_5Cl](MnO_4)_2$ | **2** |
| $[Co(NH_3)_6](MnO_4)_3$ | **3** |
| $[Co(NH_3)_6]Cl_3$ | **4** |
| $[Co(NH_3)_6]Cl_2(ClO_4)$ | **5** |
| $[Co(NH_3)_6](ClO_4)_3$ | **6** |

The blackish purple blocks of compound **1** show some birefringence (red, brown). It decomposes in water easily and explodes on fast heating with an evolution of ammonia [31]. We repeated this experiment, although the yield was quite low (16.4%) due to the solubility of compound **1** in water at room temperature (7.89 g/100 mL). Compound **1** is insoluble in aliphatic and aromatic hydrocarbons, acetone, and chlorinated solvents such as $CCl_4$, chloroform or dichloromethane, but it is soluble in DMF (0.848 g/100 mL) and decomposes in DMSO immediately. It also decomposes in wet state in a day but when it is dry and in the absence of light, it can be stored for several days. Its powder X-ray diffractogram confirmed the phase purity (the PXRD patterns completely agree with the peak positions calculated from the single crystal X-ray measurements, as shown in ESI Figures S1 and S2).

We tested some other preparation possibilities for compound **1**, e.g., the reaction of [hexaamminecobalt(III)] chloride with $KMnO_4$ at various molar ratios, but instead of an increased yield, a complex product $K[Co(NH_3)_6]Cl_2(MnO_4)_2$ containing $K^+$ [31] was obtained. Using $NaMnO_4$, the solubility of which is higher by one order of magnitude than the solubility of $KMnO_4$ [32], resulted in a similar product containing $Na^+$. Other alternative reaction routes to prepare permanganate salts [33–35] have also been tested, but complicated reaction mixtures formed due to side reactions involving chloride ions and the hydrolysis of compound **1**. The reaction of $([Co(NH_3)_6]Cl_2)_2SO_4$ with barium manganate would ensure an easy way to prepare compound **1**, but only the mixed chloride sulfate salt of the hexaamminecobalt(III) cation $([(Co(NH_3)_6]Cl(SO_4))$ is known [36], and the requested $([Co(NH_3)_6]Cl_2)_2SO_4$ has not been prepared yet.

Alvisi performed a series of experiments to prepare the perchlorate analog, $[Co(NH_3)_6]$ $Cl_2(ClO_4)$ (compound **5**) in the reaction of compound **4** and $NH_4ClO_4$, but only the complex $[Co(NH_3)_6]Cl(ClO_4)_2$ could be isolated [37]. Similarly, when $[Co(NH_3)_6](ClO_4)_3$ (compound **6**) reacted with hydrochloric acid, only [hexaamminecobalt(III)] chloride diperchlorate was obtained. If the hydrochloric acid was fuming, compound **4** was formed. The reaction between silver perchlorate and compound **4** also failed [37]. The analogous reactions with permanganate derivatives cannot be performed due to the reaction between hydrochloric acid and permanganate ions and the low solubility of $AgMnO_4$ [32].

### 2.2. Structure of Compound 1

Red platelet single crystals of compound **1** were selected from the mother liquor, which was formed during the synthesis and measured by the single crystal X-ray diffraction method. Selected crystallographic data based on the refinement results are listed in Table 2. The structural features of compound **1** are given in Figures 1–4. The detailed structural parameters, bond lengths and angles including hydrogen bond parameters are given in Tables S1–S3. The powder X-ray data calculated from the single crystal XRD results agreed very well with the powder XRD results found experimentally (ESI Figures S1 and S2).

**Table 2.** Crystal data of compound **1**.

| **Empirical Formula** | $[Co(NH_3)_6]Cl_2(MnO_4)$ |
|---|---|
| Formula weight | 350.97 g·mol$^{-1}$ |
| Crystal system | Monoclinic |
| Space group | $P2_1/c$ |
| Unit cell dimensions, Å | $a = 13.6133\ (7)$ $b = 7.3658\ (5)$ $c = 12.3682\ (6);$ $\beta = 108.547\ (8)°$ |
| Z | 4 |
| Density (calcd.) (g·cm$^{-3}$) | 1.983 |
| Temperature (K) | 163 |
| Volume (Å$^3$) | 1175.78 (13) |
| R factor (%) | 4.22 |
| CSD deposition number | 2,220,607 |

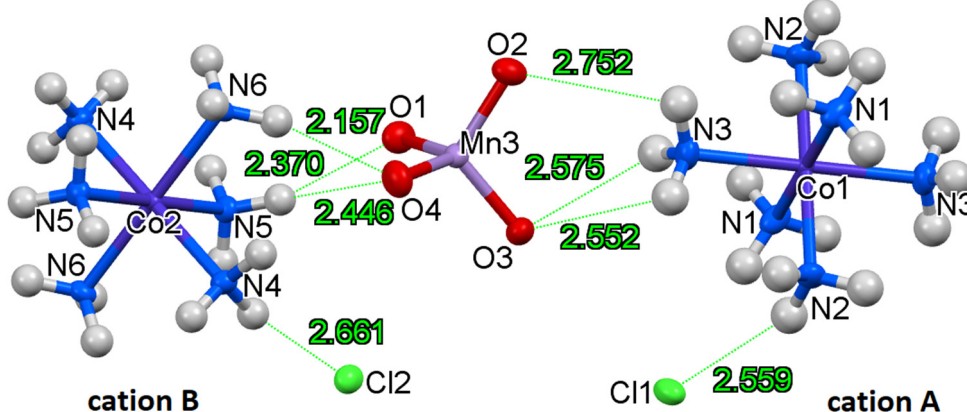

**Figure 1.** Hydrogen bond interactions and atom labeling in the structure of hexaamminecobalt (III) dichloro permanganate (atomic displacement parameters are drawn at the probability level of 50%).

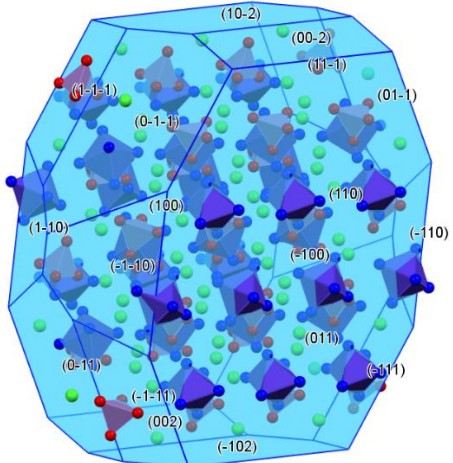

**Figure 2.** BDFH-predicted morphology of compound **1** calculated by Mercury software showing the {−100} facet in front (hydrogen atoms were omitted for clarity).

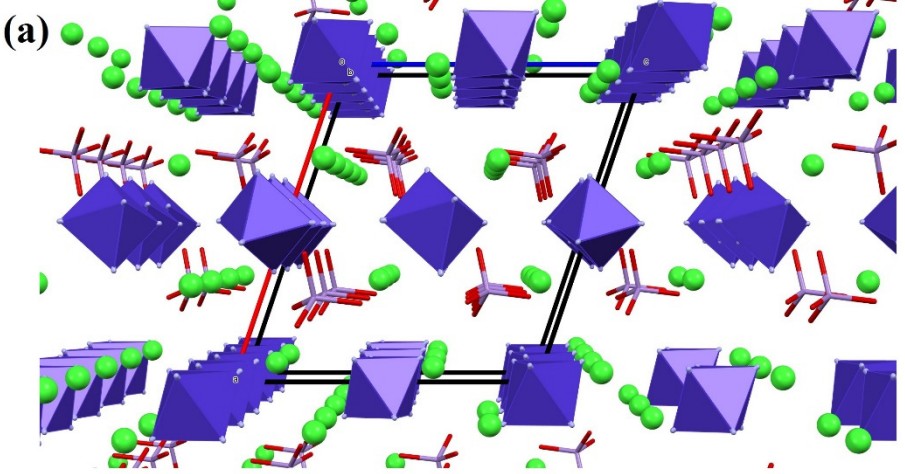

**Figure 3.** *Cont*.

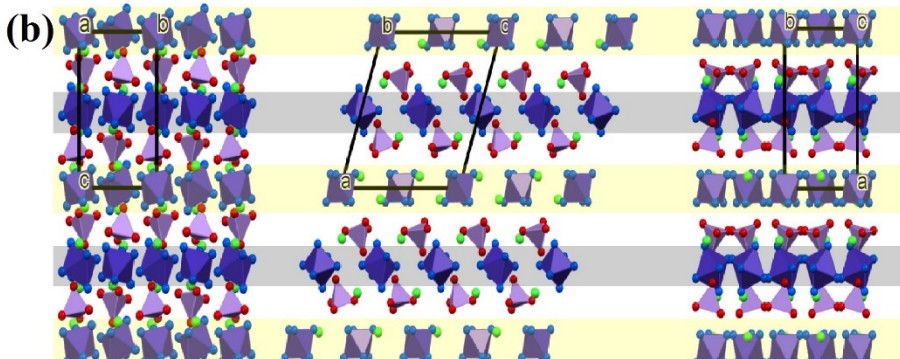

**Figure 3.** (**a**) Packing arrangement in the crystal of hexaamminecobalt (III) dichloro permanganate. (**b**) Packing arrangement in the crystal of compound **1** from various directions. Different cation layers are indicated by colored rectangles. Layers of cation *A* are indicated by yellow and layers of cation *B* are indicated by gray (the structure is drawn in stick representation).

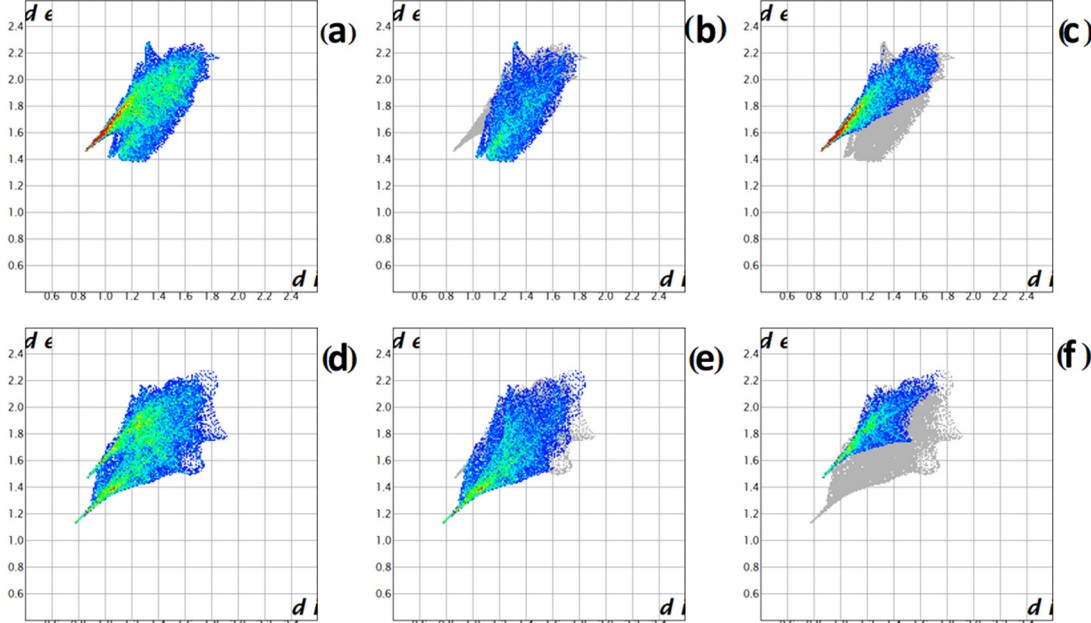

**Figure 4.** Fingerprint plots for the two hexaamminecobalt cations ((**a**) intermolecular interactions of cation *A*, (**b**) N–H⋯O interactions of cation *A*, (**c**) N–H⋯Cl interactions of cation *A*, (**d**) intermolecular interactions of cation *B*, (**e**) N–H⋯O interactions of cation *B*, (**f**) N–H⋯Cl interactions of cation *B*).

Compound **1** crystallized in the monoclinic system in the $P2_1/c$ (Nr. 14) space group. The asymmetric unit contains two halves of the hexaamminecobalt(III) complex cation, two chloride anions and one permanganate anion, whereas the unit cell contains four hexaamminecobalt (III) dichloride permanganate complexes. There are two different complex cations with a somewhat distorted octahedral geometry (bond angles ranging between 88.1° and 91.9°) (Figure 1). The two different cations (labeled as *A* and *B*) are hydrogen bonded to the permanganate oxygens; with different geometries; the complex cation *B* has considerably shorter hydrogen bond lengths.

No direct metal–metal interactions were found in the structure; the shortest Co–Co, Co–Mn and Mn–Mn distances are 7.198 (1) Å, 5.011 (1) Å, and 6.895 (1) Å, respectively. The arrangement of metallocenter polyhedra are given in Figure 2.

The packing arrangements along the directions of crystallographic axes *a*, *b* and *c* are shown in Figure 3. Two types of cationic layers can be found in the structure. In the first

type of layer, cation *A* is placed together with Cl1 anions, whereas cation *B* is placed in a different type of layer without any chloride ions. The $Cl_2$ ion is pushed into the anionic layer formed by the permanganate anions. The permanganate layers are in very close contact with the cation *B* layers, and they are moderately further from the cation *A* layers where the $Cl_1$ anions decrease the positive charge.

A high number of rather strong hydrogen bond interactions exist between the complex cations and both types of anions (a total of 25 hydrogen bonds). Each ammonia forms three to five hydrogen bonds with the anions. $Cl_1$ accepts six hydrogen bonds, and $Cl_2$ accepts five hydrogen bonds. The hydrogen bonds that formed with the chloride anions are on average weaker than those established by the permanganate anions. The strongest hydrogen bonds are between cation *B* and the permanganate ion. The hydrogen bonds of cation *A* are somewhat longer and the $D-H\cdots A$ angles are on average less favorable. A permanganate anion is fixed by 14 hydrogen bonds in the crystal lattice.

To explore the intermolecular interactions of the two crystallographically independent cations, we performed a Hirshfeld surface analysis by partitioning the space within a crystal structure into regions, in which the electron density from a sum of atoms of the given molecule dominates over the sum of the electron densities of the crystal. Several 2D fingerprint plots were generated for the complex cations (Figure 4). In the fingerprint plots, $d_e$ is represented against $d_i$ where $d_i$ is the distance from the Hirshfeld surface to the nearest atom internal to the surface and $d_e$ is the distance of the surface to the nearest atom external to the surface.

The fingerprint plots for the two cations show marked differences. The strongest $H-$bonds are formed with cation *B*, which is indicated by the much lower $d_e$ and $d_i$ values (Figure 4d). For cation *B*, two independent spikes can be seen on the plot: one for the N–H$\cdots$O and one for N–H$\cdots$Cl interaction (Figure 4e,f). The spike of the N–H$\cdots$Cl interaction appears at much higher $d_e$ values. For cation *A*, the spike for the N–H$\cdots$O interactions is missing (Figure 4b), which shows that the interactions of cation *A* with the permanganate are slightly loose (the same could also be concluded from the $H-$bond interaction lengths in ESI Table S2). This is a marked difference from cation *B*, where the N–H$\cdots$O spike is pronounced (Figure 4e). The number of the N–H$\cdots$Cl interactions of cation *A* is much higher than for cation *B* (Figure 4c), as indicated by red in the fingerprint plots while the interaction distances; thus, the shape of the spikes are similar.

The strong asymmetry of the hydrogen bond interactions of the ammonia ligands of cation *A* and *B* may play a key role in the occurrence of the selective ammonia oxidation reaction, leading to the oxidation of only a part of ammonia ligands into nitrate (see below).

### 2.3. Spectroscopic Properties of Compound **1**

We analyzed the vibrational spectra (IR and Raman) of compound **1** by means of factor group analysis and the available spectroscopic data of $[Co(NH_3)_6]^{3+}$ cation [38–42]. The structure of $[Co(NH_3)_6]Cl_2(MnO_4)$ in the unit cell can be considered composed of a $Co^{III}$ ($C_i$) and two chloride ($C_1$) ions, six $NH_3$ molecules ($C_1$), and one $MnO_4^-$ anion ($C_1$), taking into consideration that there are six crystallographically different $NH_3$ molecules ligated to two crystallographically different half$-Co^{III}$ centers.

As mentioned above, compound **1** is monoclinic ($P2_1/c$, $Z = 4$). There are a total of 36 internal permanganate vibrational modes: nine of each symmetry species. Those are one $\nu_1$ ($\nu_s$) mode, a doublet of the $\nu_2$ ($\delta_s$) mode, and two triplets due to $\nu_{as}$ and $\delta_{as}$ modes, respectively, expecting up to 18 bands ($9A_u$ and $9B_u$) in the IR and the same number ($9A_g$ and $9B_g$) in the Raman spectra (Figure 5). Altogether, 12 hindered rotational and 12 hindered translational modes are expected in the IR ($6A_u$ and $6B_u$) and Raman ($6A_g$ and $6B_g$) spectra (Figure 5), respectively.

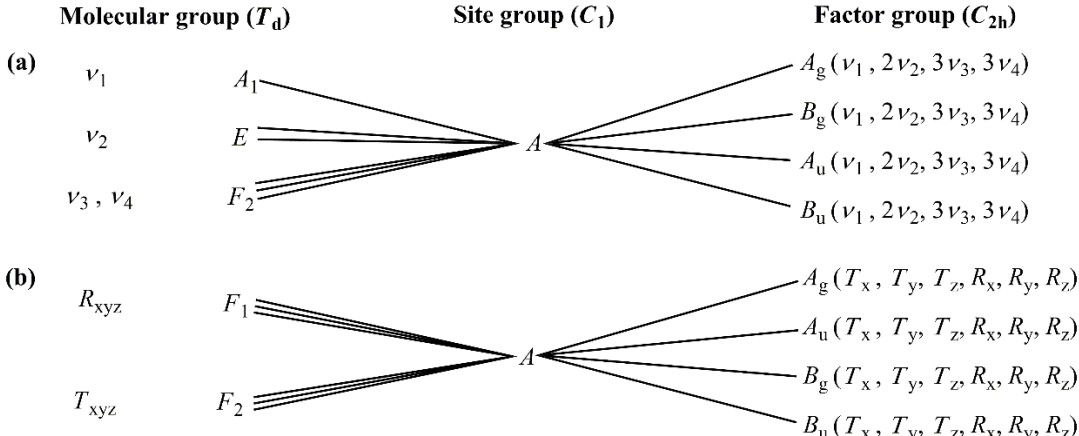

**Figure 5.** (**a**) Internal and (**b**) external permanganate vibrations in compound **1**. $\nu_1$—symmetric stretch; $\nu_2$—symmetric bend; $\nu_3$—antisymmetric stretch; $\nu_4$—antisymmetric bend.

For two crystallographically different types of Co atoms, the number of translational modes is doubled ($2 \times 6$) ($2 \times 3$ in $A_u$ and $2 \times 3$ in $B_u$) (ESI Figure S3). For two crystallographically different types of Cl atoms at positions of trivial symmetry, the number of modes is also doubled ($2 \times 12$). Namely, six hindered translations are expected of each symmetry, giving 12 bands in the IR and in the Raman spectra each, respectively (ESI Figure S4).

There are six different crystallographic types of $NH_3$ molecule at the position of trivial symmetry, $C_1$. Since 12 modes are expected in the IR ($6A_u$ and $6B_u$) and another 12 are expected ($6A_g$ and $6B_g$) in the Raman spectra ($1-1$ $\nu_1$ ($\nu_s$), $1-1$ $\nu_2$ ($\delta_s$) and $2-2$ $\nu_3$ ($\nu_{as}$) and $2-2$ $\nu_4$ ($\delta_{as}$)) for one crystallographic type of ammonia ligands, the total number of internal vibrations are $6 \times 24 = 144$ (Figure 6). Translations along or rotations around an axis are presented by lower indices of the axis/axes in question. For six crystallographic types of $NH_3$ molecules, the number of modes is correspondingly six times larger, i.e., $6 \times 12 = 72$ hindered rotations and 72 hindered translations.

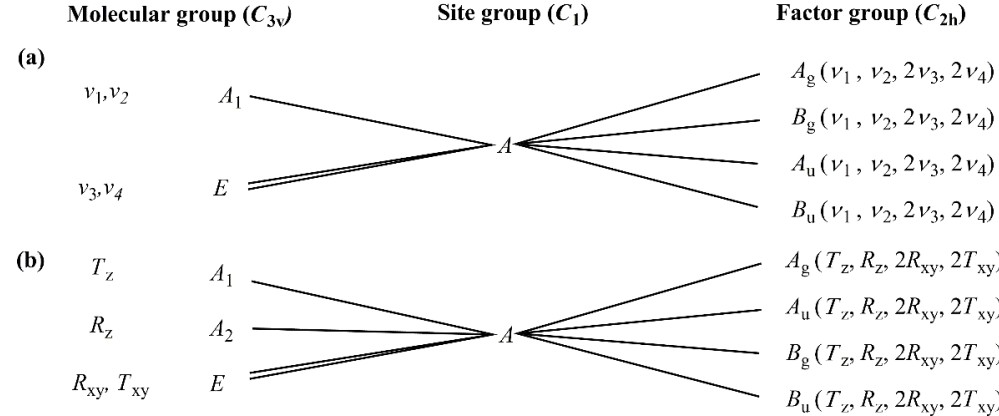

**Figure 6.** (**a**) Internal and (**b**) external ammonia vibrational modes in compound **1**. $\nu_1$—symmetric stretch; $\nu_2$—symmetric bend; $\nu_3$—antisymmetric stretch; $\nu_4$—antisymmetric bend.

For the 32 atoms in the formula unit multiplied by 4 (value of $Z$ in the primitive cell) and multiplied by 3 ($3N$, where $N = 32 \times Z = 128$), the total number of rotational degrees of freedom (hindered rotations) is altogether 84, and that of the internal vibrations is 180. The total number of hindered translations in compound **1** is 120 (72 for the ammonia ligands and 48 for other parts of the complex). Three of them belong to acoustic modes, and the rest (117) are vibrations of translational origin. These give a total number of 384 degrees of freedom.

### 2.3.1. Vibrational Modes of the Permanganate Anion in Compound **1**

The IR and Raman spectra of compound **1** are given in Figure 7 and ESI Figures S5–S9, and the band assignments can be seen in Tables S4–S6. Two series of vibrational modes of the permanganate ions (singlet symmetric stretching ($\nu_1$), triplet antisymmetric stretching ($\nu_3$) and bending ($\nu_4$), or doublet symmetric deformation ($\nu_2$)) are expected to appear in both the IR and Raman spectra (all four normal modes of tetrahedral permanganate ions are Raman active, and the IR forbidden $\nu_1$ and $\nu_2$ are also expected to appear due to the distortion from the ideal tetrahedral symmetry (Figure 5)).

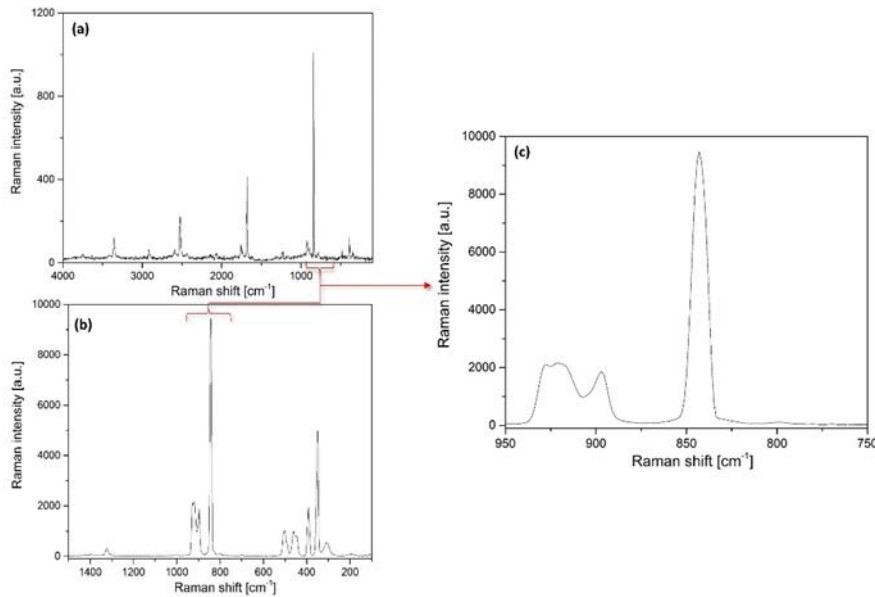

**Figure 7.** Raman spectra of compound **1** with (**a**) 532 and (**b**,**c**) 785 nm excitation.

The stretching modes of the permanganate ion in compound **1** appear as a weak singlet (852 cm$^{-1}$, $\nu_s$) and as a doublet with a shoulder (924 sh, 910 and 894 cm$^{-1}$, $\nu_{as}$). In the Raman spectra measured at room temperature and at −123 K, however, the antisymmetric stretching mode appears as four bands and a shoulder (Figure 7, ESI Table S4). The band belonging to the $\nu_s$ mode is weak in the IR spectra, whereas it is very strong in the Raman spectra. The antisymmetric stretching mode is the strongest band in the IR spectra; however, it is medium intensity in the Raman spectra. The deformation modes of the permanganate ion are located in the far-IR range. $\delta_s$ cannot be seen due to its forbidden nature under $T_d$ in the IR spectra, whereas a weak and poorly resolved triplet of $\delta_{as}$ appears near 388 cm$^{-1}$. A singlet and a singlet with a shoulder appear both in room-temperature and low-temperature Raman spectra of compound **1** for $\delta_s$ and $\delta_{as}$ modes, respectively.

The Raman measurement with 532 nm excitation resulted in resonance Raman effects [43] (Figure 7), and a series of overtone bands appeared at 846, 1676, 2526, and 3354 cm$^{-1}$ ($\nu_s$, $2\nu_s$, $3\nu_s$ and $4\nu_s$) with decreasing intensity. The bands of antisymmetric stretching mode ($\nu_{as}$, $2\nu_{as}$, $3\nu_{as}$ and $4\nu_{as}$) as weak bands also appeared (Figure 7). Although $\nu_2$ (Co–N is IR inactive under $O_h$, this mode might be mixed with the $\delta_{as}$ (Mn–O) due to the distortion of regular octahedral structure in compound **1**.

### 2.3.2. Vibrational Modes of the [Hexaamminecobalt(III)] Cation in Compound **1**

The correlation analysis of the cationic part of compound **1** showed two sets of vibrational modes belonging to $2 \times 6$ different ammonia molecules and two octahedral $CoN_6$ skeletons. Based on the normal coordinate analysis and band assignations of the hexaamminecobalt(III) cation [38–42], the assignments of the cationic vibrational modes of compound **1** are given in ESI Tables S5 and S6.

The six crystallographically different ammonia molecules in each cation type (*A* and *B*) resulted in poorly resolved complex band systems for each N–H mode. The symmetric and antisymmetric deformation modes appear around 1340 cm$^{-1}$ (with a shoulder at 1327 cm$^{-1}$) and as a wide asymmetric band at 1608 cm$^{-1}$, respectively. The appearance of the shoulder that belongs to the symmetric deformation mode $\nu_s$ (N–H) of compound **1** showed that the 2 × 6 different ammonia ligands can be divided into at least two sets ligated with significantly different strengths to the central Co$^{III}$−ion. A relative bond strength parameter ($\varepsilon$) for the ammonia molecules in ammine complexes was defined by Grinberg [11,44]. The parameter ($\varepsilon$) was found to be 0.94 and 0.90 for the two groups of the coordinated ammonia ligand types in compound **1**. The bond strength difference between these groups of coordinated ammonia ligands in compound **1** is only ~4%, and it might be attributed to the differences between the Co–N bond strengths in the apical or equatorial positions or between the cations of type *A* and *B*. Among the vibrational modes belonging to the ammonia ligand, only the rocking mode $\rho(NH_3)$ is sensitive enough to characterize the strength of hydrogen bonds in ammonia complexes [39]. This shows that the average strength of the hydrogen bonds in compound **1** is somewhere between the strength of an average hydrogen bonds in [Co(NH$_3$)$_6$]Cl$_3$ ($\rho(NH_3)$ = 830 cm$^{-1}$) [39] and that in [Co(NH$_3$)$_6$](MnO$_4$)$_3$ ($\rho(NH_3)$ = 803 cm$^{-1}$) [45].

Only the $\delta_s$(NH) and $\rho(NH_3)$ Raman bands were visible at room and liquid N$_2$ temperature as a wide and a weak band consisting of three components (ESI Table S5).

The CoN$_6$ octahedron under O$_h$ has six normal modes, among which $\nu_1(\nu(CoN)$, A$_g$), $\nu_2(\nu_{as}$, E$_g$) and $\nu_5(\delta_s$, F$_{2g}$) are Raman, whereas $\nu_3(\nu_s$, F$_{1u}$) and $\nu_4(\delta_{as}$, F$_{1u}$) are only IR active modes. The $\nu_6(\delta(NCoN)$, F$_{2u}$) mode is IR and Raman inactive mode. These band positions were calculated by normal coordinate analysis methods [40–43]. The $\nu_1$ mode has a singlet nature; thus, the two bands in the Raman spectra probably belong to separated $\nu_{Co–N}$ modes of two different Co–N moieties. An intense band consisting of $\nu_3$(Co–N)($\nu_s$) and $\nu_4$(NCoN) ($\delta_{as}$) appears in the IR spectrum, whereas all the $\nu_1$–$\nu_5$ modes can be found in the Raman spectra due to the distortion of the regular CoN$_6$ octahedron. The splitting of the bands belonging to the $\nu_2$–$\nu_5$ modes can be attributed to the presence of various Co–N distances in the two different CoN$_6$ skeletons (ESI Figure S7), and at the same time, to the removal of the degeneracy of E and F levels. The geometry distortion results in the appearance of the forbidden $\nu_6(\delta_{NCoN})$ band as well as a shoulder in the IR spectrum around 250 cm$^{-1}$.

### 2.4. UV−VIS Spectroscopy

The UV−VIS spectra of solid compound **1** was recorded at room temperature (ESI Figure S10). The spectrum consists of strongly overlapping bands of four possible *d*–*d* transitions of the [Co(NH$_3$)$_6$]$^{3+}$ cation and CT bands of the permanganate anion [8,46]. As a low-spin complex cation, the ground state of [Co(NH$_3$)$_6$]$^{3+}$ is t$_{2g}^6$ ($^1$A$_{1g}$). The electron is excited, and the t$_{2g}^5$e$_g$ excited state spans with $^3$T$_{1g}$ + $^1$T$_{1g}$ + $^1$T$_{2g}$ + $^3$T$_{2g}$ terms. The triplet states lie at lower energies than the singlet states. The intensity of spin-allowed transitions (singlet terms) was expected to be weak [47–51]. The presence of hydrogen bonds with water in aqueous solutions resulted in trigonal distortion of an octahedral structure and the appearance of new bands [48]. The experimentally found UV−VIS data are given in ESI Table S7.

The $^1$A$_1$→$^1$T$_1$ and $^1$A$_1$→$^1$T$_2$ transitions of the octahedral Co$^{III}$ cation are spin-allowed. The distortion due to hydrogen bonds results in trigonal distortion (compression), which was found in the experimental electronic spectrum of the aq. solutions of [Co(NH$_3$)$_6$]Cl$_3$ and also showed by DFT calculations with water around the [Co(NH$_3$)$_6$]$^{3+}$ cation [48]. The band observed at 250 cm$^{-1}$ may be assigned both to the CT band of the complex cation and the $^1$A$_1$−$^1$T$_2$ (3t$_2$−2e) transition of the permanganate ion (it was found at 259 nm for KMnO$_4$), whereas the band at 220 nm may be assigned to the $^1$A$_1$−$^1$T$_2$ (t$_1$−4t$_2$) transition of a permanganate ion (it was found at 227 nm for KMnO$_4$) [8].

In the visible region of spectra, the bands at 510 and 530 nm belong to the permanganate $^1A_1-^1T_2$ ($t_1-2e$) transition, whereas the bands at 490 and 551 nm may belong to the permanganate and cation transitions (ESI Table S7) as well. A similar band system was found in the UV−VIS spectrum of $KMnO_4$ between 500 and 562 nm [8]. The band at 725 nm is the strongest band and probably consists of the $^1A_1-^1T_1(t_1-2e)$ transition of the permanganate ion and a weak component of the $^1A_1\rightarrow^5T_2$ transition of the complex cation. The $^1A_1-^1T_1(t_1-2e)$ transition of the permanganate ion was found at 720 nm for $KMnO_4$ and 710 nm for $[Agpy_2]MnO_4$ [8].

### 2.5. Non-Isothermal Thermal Decomposition of Compound **1**

Compound **1** is not thermally stable and behaves as an explosive on heating; therefore, its decomposition had to be performed with a low heating rate (2 °C min$^{-1}$) until 150 °C to avoid an explosion-like decomposition. The temperatures of the first DTG peak of compound **1** recorded under an inert atmosphere and an atmosphere with oxygen content were 107 and 129 °C, respectively (Figure 8a,b). The DSC peak temperatures, however, were the same in $O_2$ and $N_2$ atmospheres (109 and 134 °C for the 1st and 2nd decomposition steps in both atmospheres, respectively) (Figure 8c,d), whereas the reaction heats were found to be different (−107.1 and −260.8 kJ/mol in the first and −90.3 and −64.5 kJ/mol in the second decomposition step, in $O_2$ and $N_2$ atmospheres, respectively (Figure 8)). It shows that outer oxygen does not take part directly in the starting of the decomposition reaction; however, an indirect influence can be found during the reaction, e.g., via consumption of the primary thermal decomposition products in consecutive secondary endothermic reactions. We can assume that the primary thermal decomposition products containing nitrogen in some reduced species form are oxidized in the presence of $O_2$. The formation of endothermic nitrogen compounds such as NO or $N_2O$ can explain why the reaction heat in the first decomposition step is lower in $O_2$ than in $N_2$. Furthermore, other endothermic reactions such as increased ammonia ligand loss can cause similar results.

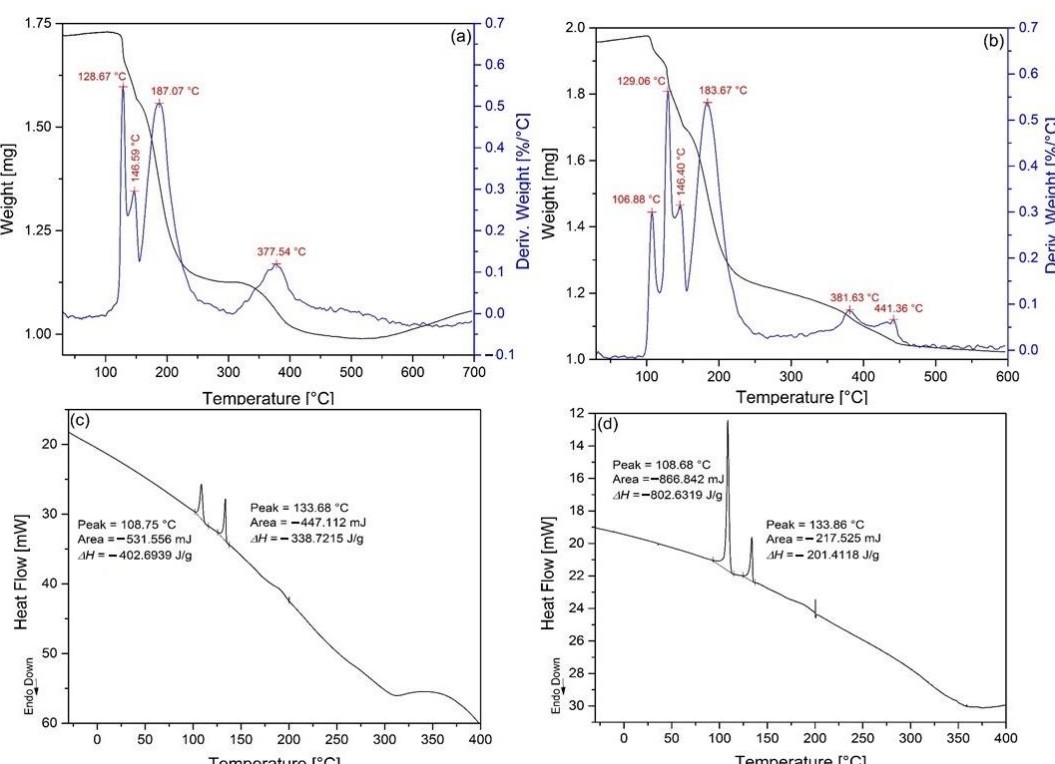

**Figure 8.** The TG and DTG curve of compound **1** in (**a**) air and (**b**) in an inert atmosphere. The DSC curve of compound **1** in (**c**) air and (**d**) in an inert atmosphere.

Under $N_2$, two other DTG peaks were observed at 382 and 441 °C, whereas in air, the oxidizable residues were completely eliminated even at 378 °C, and there was no DTG peak above this temperature. Above 500 °C, in air, the low-valence metal ion components of the thermal decomposition products that formed as reduced components (e.g., $Co^{II}$, $Mn^{II}$, $Mn^{III}$) due to redox reactions may have been oxidized in air (with the formation of $Co^{III}$, $Mn^{IV}$), which caused a weight increase due to oxygen uptake.

Since the formation of $N_2$ and $O_2$ as decomposition products during the analysis of the evolved gas, the TG−MS measurements were not performed in air but only in argon ($m/z = 40$) as an inert atmosphere (Figure 9). The TG−MS curves unambiguously show that the first three decomposition steps consist of redox reactions, because $H_2O$ ($m/z = 18$), $NO^+$ ($m/z = 30$) and $N_2O^+$ ($m/z = 44$) as redox products were detected. Compound **1** is anhydrous; thus, water may only be the oxidation product of the only possible hydrogen source—ammonia ligands—and the only possible oxygen source may be the permanganate ions. Thus, the first redox reaction is involved with ammonia ligands and permanganate ions. Water and $N_2$ formed in all three decomposition steps, $N_2O$ formed only in the first and third steps, and NO formed in the second and third steps. It shows that $NO^+$ would only be the fragment ion of $N_2O^+$. The signal at $m/z = 17$ may belong to both $OH^+$ as water fragments and an ammonia molecular ion ($NH_3^+$) as well. The $m/z = 17$ and 16 ion intensity curves indicate the release of ammonia ($NH_3^+$ and $NH_2^+$) as $m/z = 17$, and 16 fragment ions of $H_2O$ were subtracted from these curves. The intensity ratio of the $m/z = 16$ ($O^+$) fragment from water comparing to the intensity of $m/z = 17$ and 18 signals in the TG−MS of water [52] confirm unambiguously that ammonia is also present in the system. There was no peak at $m/z = 32$ ($O_2^+$) as the parent ion for $m/z = 16$ ($O^+$), and we could not detect HCl or $Cl_2$ due to a reaction of these reactive species with the wall of the capillary column in the TG−MS instrument [9] (Figure 9).

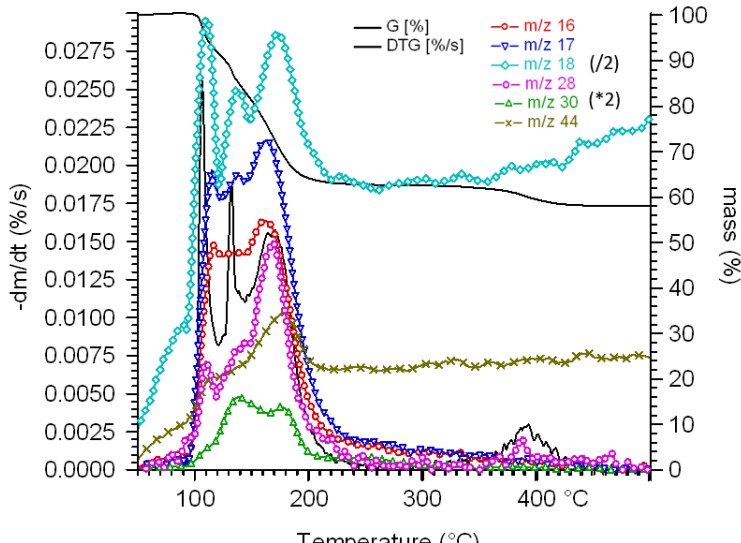

**Figure 9.** The TG−MS ion curves of compound **1** in an argon atmosphere. Please note that the intensity of the $m/z = 18$ and 30 is multiplied.

In order to clarify the nature of intermediate products and gain an insight into the reaction mechanism, we studied isotherm thermal decompositions of compound **1** in solid phase under air and $N_2$ at various temperatures that were determined on the basis of the DTG curves (Figure 8a,b) and under refluxing toluene (boiling point is 110 °C).

Toluene acted as a heat-absorbing medium preventing local overheating due to exotherm redox reactions, and it limited the decomposition temperature to 110 °C. The reaction temperature could not exceed the boiling point (110 °C) of toluene until liquid toluene is present; thus, we can study in detail the first decomposition step of compound **1** that occurred around 107 °C.

### 2.6. Isothermal Heat Treatments of Compound 1

The powder X-ray diffractograms and IR/far−IR spectra of the decomposition intermediates and products of compound **1** were recorded on the samples made by heating compound **1** for 2 h near the DTG peak temperatures and 500 °C (Figure 8a,b), respectively. The powder XRD of the decomposition products made at 500 °C showed the presence of a $(Co,Mn)^{T-4}(Co,Mn)^{OC-6}_2O_4$ phase with a Co:Mn ratio of 1:1 and an average size of ~9 nm (determined by the Scherrer method (Figure 10)).

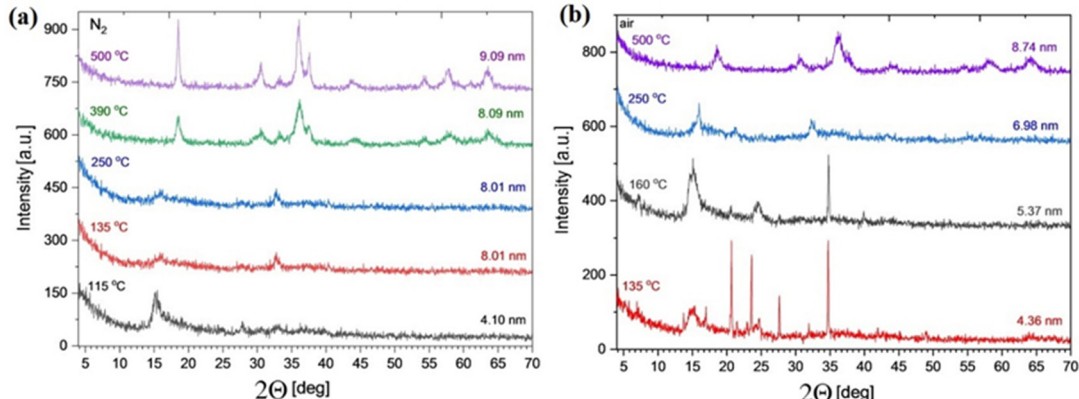

**Figure 10.** The powder X-ray diffractograms of the decomposition intermediates and products of compound **1** in (**a**) an inert atmosphere and (**b**) air.

The $Co_{1.5}Mn_{1.5}O_4$ formula determines 2 $M^{III}$ and one $M^{II}$ ions, so at least 0.5 Co is surely trivalent, even if all the manganese is trivalent, but the distribution and the valences of Mn and Co at the tetrahedral and octahedral sites may vary. A comparison of the powder XRDs of $Co_{1.5}Mn_{1.5}O_4$ with the diffractograms of $CoMn_2O_4$ (tetragonal, $I4_1/amd$) [53] and $MnCo_2O_4$ (cubic, $Fd3m$) unambiguously showed that $Co_{1.5}Mn_{1.5}O_4$ is isostructural with the cubic $Mn^{II}Co^{III}_2O_4$ [54].

The distribution of valences ($Co^{II/III}$ and $Mn^{II/III/IV}$) and distribution of the metal ions between the tetrahedral and octahedral sites of the spinel lattice strongly depend on the composition and preparation method of the Co–Mn–spinel [54–62]. Generally, cubic spinels with inverse spinel structures form when more than the half of metal ions is cobalt, whereas the manganese-rich spinel are tetrahedral. $Co^{III}$, $Mn^{III}$ and $Mn^{IV}$ favor the octahedral sites of spinel lattice, whereas $Co^{II}$ and $Mn^{II}$ have no preference. Since the redox reactions between $Co^{II}$ and $Mn^{III}/Mn^{IV}$ can result in charge re-distribution, divalent ions can form at the octahedral sites as well, especially if the synthesis temperature is low and the diffusion rate is limited [54–62].

The PXRDs of the intermediate phases produced at 135, 160, 250 and 390 °C under $N_2$ show that amorphous materials are formed until 250 °C, and the crystalline spinel structure was found to build up at 390 °C (average crystallite size is ~8 nm) (Figure 10). It is confirmed by the far−IR spectra (Figure 11) of the decomposition intermediates, where the low-frequency metal–oxygen and lattice modes characteristic of the spinel structures [26–28] appear in the spectrum of sample produced at 390 °C. Since we found no differences between the far−IR band positions of the intermediates prepared in air and an inert atmosphere between 135 and 500 °C, the development of the cobalt-manganese oxide framework looks to be independent of the composition of the atmosphere. Therefore, the role of oxygen in the thermal decomposition of compound **1** may not be associated with the build-up of these phases. Consequently, the unidentified crystalline product that appears in the XRD of the samples produced in the presence of oxygen at 135 °C is not a cobalt manganese oxide.

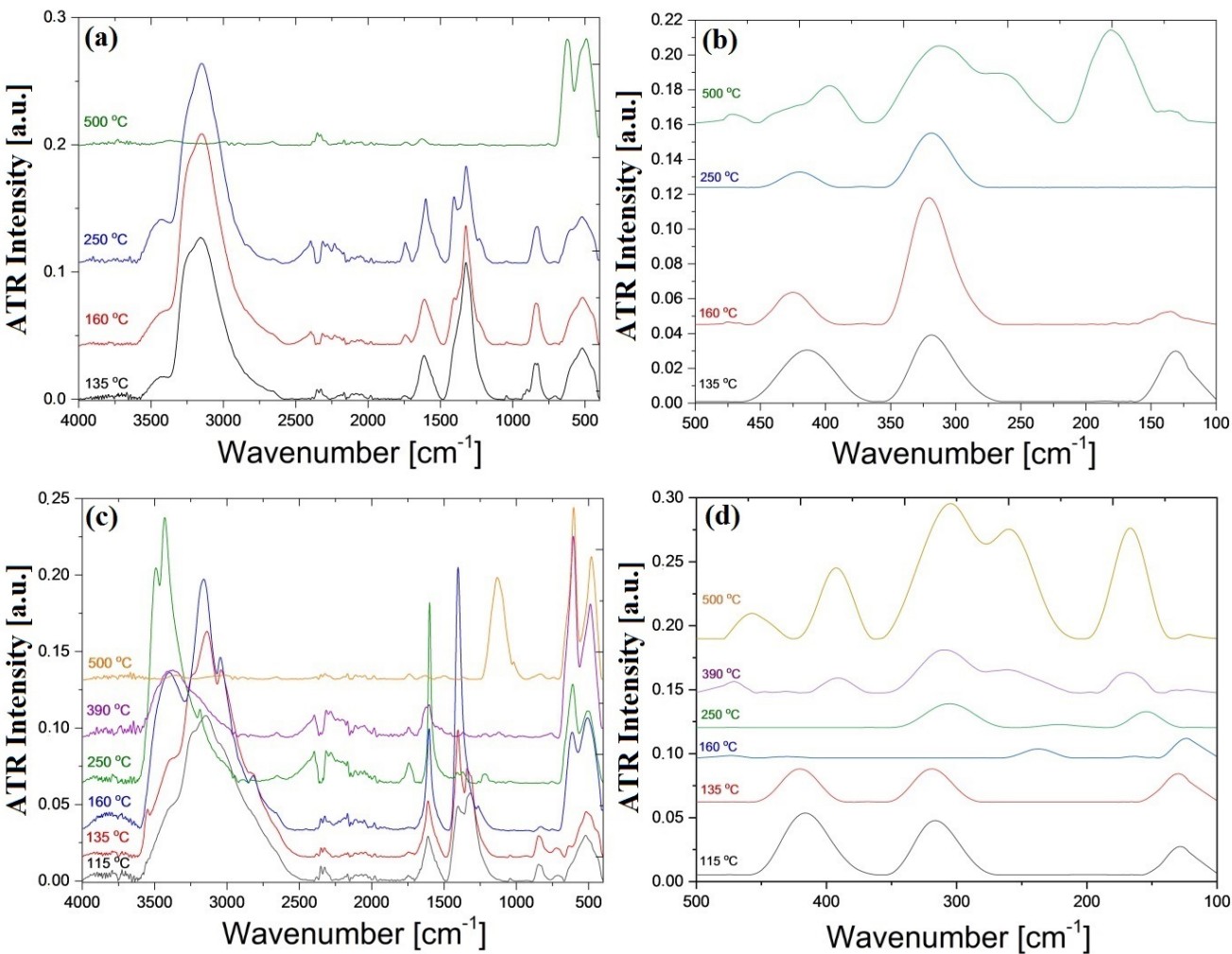

**Figure 11.** The IR/far−IR spectra of the decomposition intermediates and products of compound **1** in (**a**,**b**) air and (**c**,**d**) an inert atmosphere.

The strong antisymmetric stretching bands of $MnO_4^-$, $MnO_4^{2-}$ and $MnO_4^{3-}$ ions at ~910, ~862 and 770 cm$^{-1}$ [63], respectively, could not be detected in the IR spectrum of the intermediate that formed at 115 °C. Thus, all the permanganate ion content of compound **1** is reduced at the first moment of thermal decomposition into Mn$^{IV}$, Mn$^{III}$ or Mn$^{II}$ species. The wide band in the 600–400 cm$^{-1}$ region shows the presence of these species containing a low-valence Mn–O bond [63].

The $\nu_2$(N–O) band of the nitrate ion [64] (it may be mixed with $\rho$(NH$_3$) belonging to coordinated ammonia) appears in the samples prepared at 135, 160 and 250 °C with decreasing intensity around 850 cm$^{-1}$, which is gradually eliminated with increasing temperature. The positions of symmetric and antisymmetric N–H stretching, and deformation modes of coordinated ammonia coincide with the bands of ammonium ion N–H stretching modes, but the presence of bands/shoulders between 2900 and 2800 cm$^{-1}$ (combination bands of $\nu_2 + \nu_4$) confirms that some of the ammonia is in protonated (ammonium ion) form [65]. The intensive antisymmetric $\nu$(N–O) band of nitrate ions can be seen around 1330 cm$^{-1}$ [53], the intensity of which decreases with increasing temperature, parallel with the decrease in the $\nu_2$(N–O) band intensity. Thus, the nitrate compounds that form are not thermally stable and gradually decompose. The crystalline compound detected in the sample made at 135 °C (Figure 10) disappears at 160 °C (Figure 10), so it is thermally unstable. The formation of a phase containing water (an O–H symmetric and antisymmetric stretching band appearing at ~3500 cm$^{-1}$, and the intensity of the band containing $\delta_{as}$(N–H) and scissoring OH$_2$ components at ~1600 cm$^{-1}$ increasing as a result) starts together with

increasing ammonia/ammonium ion content (appearing of N–H symmetric and antisymmetric stretching at ~3200 cm$^{-1}$). The largest amount of products containing water were found at 250 °C, and compounds containing both water and ammonia (ammonium ion) completely disappeared at 500 °C in both atmospheres.

The IR study of the crystalline compound found in the intermediate that formed in air at 135 °C shows that a nitrate compound is present, which decomposes at 160 °C (disappearing of the $\nu_{as}$(N–O) band at 1330 cm$^{-1}$) (Figure 11). A weak stretching band of water as a shoulder at >3400 cm$^{-1}$ can also be observed, which suggests the formation of a complex containing water. When the temperature is increased, more water (increasing the scissoring $\delta$(H$_2$O) mode at ~1600 cm$^{-1}$) and less ammonia (ammonium-ion) (disappearing of the $\delta_{as}$(N–H) mode at 1400 cm$^{-1}$) are present in the IR spectrum of the sample made at 250 °C.

The reduction of Co$^{III}$ ions into Co$^{II}$ ions during the thermal decomposition of [Co(NH$_3$)$_6$Cl$_3$] and its decomposition intermediate [Co(NH$_3$)$_5$Cl]Cl$_2$ is a well-known process, similarly to other hexaamminecobalt(III) salts, with N$_2$ and NH$_4$Cl (NH$_4$X salt) formation [66,67]. Since [hexaamminecobalt(II)] complexes are thermally less stable than the analogous Co$^{III}$ complexes [68], and N$_2$ formation was detected in the thermal decomposition reaction, we performed a temperature-limited decomposition under boiling toluene. The limit of this decomposition process is 110 °C, the boiling point of toluene, which is an upper temperature limit, because the exothermic reaction heat is absorbed by the evaporation of toluene. It can prevent the decomposition of the primarily forming intermediate due to local overheating and self-propagating the decomposition reaction.

## 2.7. Isothermal Decomposition of Compound **1** in Boiling Toluene

The thermal decomposition of compound **1** starts at 110 °C, which is the boiling point of toluene. The solid phase formed in the thermal decomposition of compound **1** under refluxing toluene in 2 h consisted of a brown amorphous/hardly crystallized material, which was crystallized out—with or without washing with water before heating—at 500 °C into spinel phases with 1:1 and 1:3 Co:Mn stoichiometry, respectively (Figure 12a–d). The difference between the Co:Mn ratio with and without aqueous leaching unambiguously shows that in the first decomposition step, at least one water-soluble Co compound also forms. Therefore, the aqueous extract was evaporated to dryness, and the solid residue was studied by IR and powder XRD (ESI Figures S11 and S12). The powder XRD of the evaporation residue from the aqueous extract of the decomposition intermediate made in boiling toluene shows the presence of [Co(NH$_3$)$_6$]Cl$_3$ and [Co(NH$_3$)$_6$]Cl$_2$ as well (ESI Figure S12).

A layered compound with high interlayer distance, characteristic of basic cobalt salts might also form as a hydrolysis product ($2\theta = 14.7°$) (ESI Figure S12). We found a small amount of the cubic $\alpha-$ammonium chloride (ESI Figure S12). The IR spectrum of this residue contains a band characteristic of coordinated ammonia, ammonium ions and nitrate ions as well [9] (ESI Figure S11). These results unambiguously show that the ammonia is partially oxidized into nitrate ions, reducing the permanganate ion content completely and partly the Co$^{III}$ ions as well. Only a part of ammonia (due to the oxygen deficiency of one permanganate toward six ammonia) can be oxidized; therefore, the residual ammonia is left back as coordinated ammonia or ammonium ions. According to this, the mass decrease during the thermal decomposition in toluene roughly corresponds to the weight of ~3NH$_3$ or 2NH$_3$ + H$_2$O. The nitrate and chloride counter-ions are crystallized from the aqueous leachate as ammonium nitrate and hexaamminecobalt(III) and hexaamminecobalt(II) chlorides.

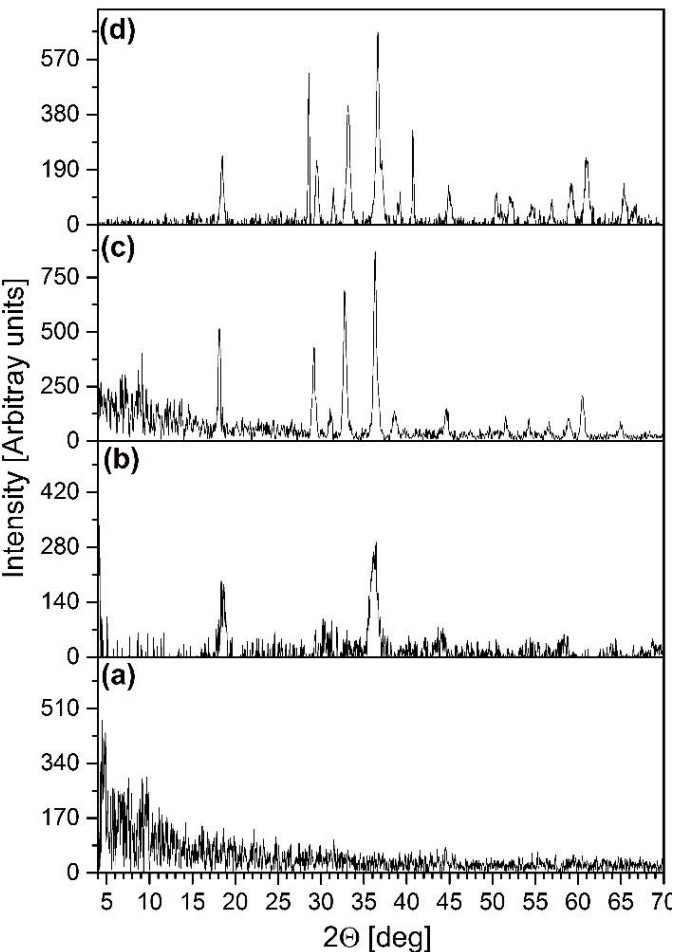

**Figure 12.** The powder X-ray diffractograms of the decomposition intermediates and products of compound **1**: (**a**) made by heat treatment in toluene at 110 °C, (**b**) made by the heat treatment of compound **1** at 500 °C in air and (**c**) made from the water-insoluble part of the decomposition intermediate of compound **1** made in toluene by heating at 500 °C and (**d**) made from the decomposition intermediate of compound **1** made in toluene by heating at 500 °C.

The water-insoluble decomposition residue contains cobalt and manganese in an atomic ratio of 1:3 (ICP). Based on the weight loss during the aqueous extraction and the IR studies (ESI Figure S13), the residue can be characterized with the formula of $Co_2(NH_3)_4Mn_6O_{12}$ or $(NH_4)_4Co_2Mn_6O_{12}$ (todorokite-skeleton [69,70] with square-hape channels intercalated with 4 ammonia or 4 ammonium ions). We found a similar phase during the decomposition of compound **2** under toluene but did not determine the chemical form of ammonia and valence distributions of the Co and Mn ions in the material that was formed (ESI Figure S14). Therefore, we now measured the room-temperature magnetic susceptibility of the todorokite-like materials forming under similar conditions from compounds **1** and **2** (5.9 B.M.).

The samples made by the decomposition of compounds **1** and **2** under toluene and after aqueous leaching may contain the four ammonia molecules as coordinated (to $Co^{II}$ or $Co^{III}$ [9]) or protonated ammonia molecules (ammonium ion). The IR band positions of $C_{3v}$ (or lower symmetry) $M \cdots NH_3$ or hydrogen-bound intercalated ammonium ion ($\cdots H^+ - NH_3$) species are quite similar, but the existence of combination bands ($\nu_2 + \nu_4$ of tetrahedral ammonium ion) in the IR spectra of these intermediates strongly suggest the presence of ammonium ions.

The magnetic susceptibility values are in accordance only with the presence of two high-spin $Co^{III}$, two high-spin $Mn^{III}$ and four high-spin $Mn^{II}$ ions. Other valence combinations of cobalt ($2Co^{II}$ or $1 Co^{II} + 1 Co^{III}$) or manganese (various numbers of $Mn^{II}$, $Mn^{III}$ or

$Mn^{IV}$ with 0 (ammonia ligands), 1–3 (some of the ammonia molecules protonated) or four ammonium ions gave much lower theoretical magnetic moments than the measured values, or they cause controversies in charge balances (the cation charges are to be neutralized by $12 \times 2 = 24$ negative charges).

Based on the IR, PXRD, magnetic susceptibility, TG and TG−MS results, the main reactions during the thermal decomposition of compounds **1** and **2** in toluene at 110 °C can be summarized with the following equations:

$$6[Co(NH_3)_6]Cl_2MnO_4 = 4Co(NH_3)_6Cl_3 + (NH_4)_4Co_2Mn_6O_{12} + NH_4NO_3 + 9H_2O + 24NH_3 + 5N_2$$

and

$$3[Co(NH_3)_5Cl](MnO_4)_2 = [Co(NH_3)_5Cl]Cl_2 + (NH_4)_4Co_2Mn_6O_{12} + NH_4NO_3 + 6H_2O + 3NH_3 + 1.5N_2$$

With and without leaching the water-soluble Co−compounds, $[Co(NH_3)_6]Cl_3$ and $[Co(NH_3)_6]Cl_2$, the heat-treatment of the decomposition intermediate that formed under refluxing toluene, at 500 °C, led to spinel phases with a Co:Mn ratio of 1:3 and 1:1 ($Co_{0.75}Mn_{2.25}O_4$, and $Co_{1.5}Mn_{1.5}O_4$), respectively. Under analogous conditions, the tetragonal Co–Mn spinel with a Co:Mn stoichiometry of 1:2 and 1:3, respectively, formed from compound **2**. Without removing the water-soluble intermediate cobalt compounds, the formal equations of the decomposition reactions at 500 °C are:

$$6[Co(NH_3)_6]Cl_2MnO_4 = 4Mn^{II}Co^{III}_{1.5}Mn^{III}_{1.5}O_4 + 2N_2O + 6H_2O + 28NH_3 + 12HCl + 2N_2$$

$$3Co(NH_3)_5Cl](MnO_4)_2 = 3CoMn_2O_4 + N_2O + 9H_2O + 8NH_3 + 3HCl + 2.5 N_2$$

After removing the water-soluble components, the formula of $(NH_4)_4Co_2Mn_6O_{12}$ determines the formation of $Co_{0.75}Mn_{2.25}O_4$ compound. Under $N_2$, atmospheric oxygen cannot help the oxidation of ammonium ions.

The 2nd and 3rd decomposition steps of compound **1** belong to the further catalytic/non-catalytic thermal decomposition of $NH_4NO_3$ and $[Co(NH_3)_6]Cl_3/[Co(NH_3)_6]Cl_2$ precursors [66–68,70–72]. The source of $N_2O$ and $H_2O$ may be ammonium nitrate; NO can be formed via the reaction of ammonia and the oxide phases. The high similarity of the decomposition pattern of compound **1** and **2** can be explained with the fact that $[Co(NH_3)_6]Cl_3$ forms during the thermal decomposition of compound **1**, which transforms during its thermal decomposition first into $[Co(NH_3)_5Cl]Cl_2$ [66,67]. $[Co(NH_3)_5Cl]Cl_2$ forms as an intermediate during the thermal decomposition of compound **2**.

The direct thermal decomposition of compound **1** and **2** at 500 °C resulted in cubic and tetragonal spinel phases, respectively. The thermal decomposition of compounds **1** and **2** in toluene, then heating at 500 °C (without washing out the water-soluble Co−compounds) gave tetragonal spinels with the same stoichiometry as that found in the simple solid-phase decomposition. The removal of water-soluble Co−compounds from the thermal decomposition products of compounds **1** and **2** in boiling toluene, however, resulted in the same product in both cases with Co:Mn = 1:3 stoichiometry ($(NH_4)_4Co_2Mn_6O_{12}$). Heat treatment of ($(NH_4)_4Co_2Mn_6O_{12}$) at 500 °C led to the same spinel phases with a tetragonal lattice (Figure 12, Scheme 1).

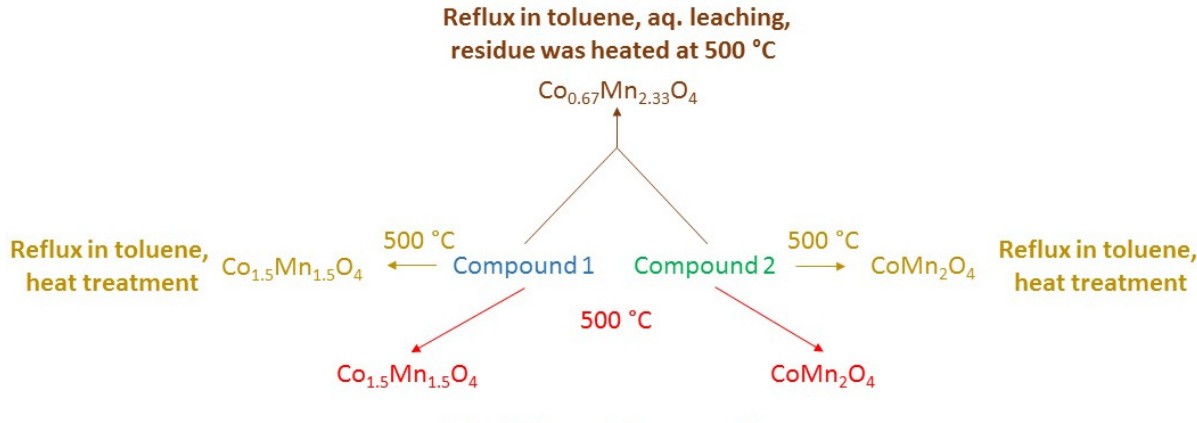

**Scheme 1.** Thermal decomposition products formed from compounds **1** and **2** at 500 °C with and without various pre-treatments.

The distribution possibility of cationic species in the spinel lattice shows high variability depending on the composition, preparation method, and synthesis conditions of the Co–Mn–spinels [55–62]. Even the normal spinel structures with $Co^{II}$ in the tetrahedral sites as $Co^{II}[Co^{III},Mn^{III}]_2]O_4$ [56,57] show high variability due to redox equilibriums between the $Mn^{II} + Mn^{IV} = 2Mn^{III}$ and $Co^{II} + Mn^{IV} = Co^{III} + Mn^{III}$ at the octahedral sites when the spinels with $Co^{II}[Co^{II}Mn^{IV}]O_4$ [58,59], $Co^{II}[(Co^{III},Mn^{II},Mn^{III},Mn^{IV})_2]O_4$ [60], $Co^{II}[(Co^{III},Mn^{III},Mn^{IV})_2]O_4$ [61], and $Co^{II}[(Co^{II},Co^{III},Mn^{IV})_2]O_4$ [61] were found. Since $Co^{II}$ and $Mn^{II}$ have no preferential site within the spinel lattice, $Mn^{II}$ can substitute $Co^{II}$ at the T−4 site, and not only the $(Co^{II},Mn^{II})[Mn^{III}_2]O_4$ spinel [56] but the $(Co^{II},Mn^{II})[Co^{II},Co^{III},Mn^{III}]O_4$ spinels [62] were also identified. This high variability in the oxidation states of each metal ion in both T−4 and OC−6 spinel positions results in the possibility of preparing promising Co–Mn–oxide catalysts for various industrially important processes. Therefore, a detailed study of the composition, metal and charge distributions, magnetic properties, and catalytic activity of Co–Mn spinels prepared from various ammine complexes of cobalt permanganates, including compounds **1** and **2**, will be studied and published in our forthcoming paper.

*2.8. Surface Characterization of the Decomposition Products Form from Compound **1** and Their Photocatalytic Activity in the Degradation of Organic Dyes*

The BET surface area determination, SEM morphological characterization and photocatalytic activity studies on the thermal decomposition intermediates and end-products formed (up to 500 °C in $N_2$ and air atmospheres) were performed.

The morphology of the end-product (spinel-like) obtained at 500 °C under $N_2$ and air show pumice-like and lamellar morphology, respectively (Figure 13). The evolution of gas with the disruption of the crystalline lattice resulted in an amorphous structure which transforms into badly crystallized spinel. Table 3 contains the specific surface area (BET, $N_2$ isotherm) of each intermediate and the final decomposition products.

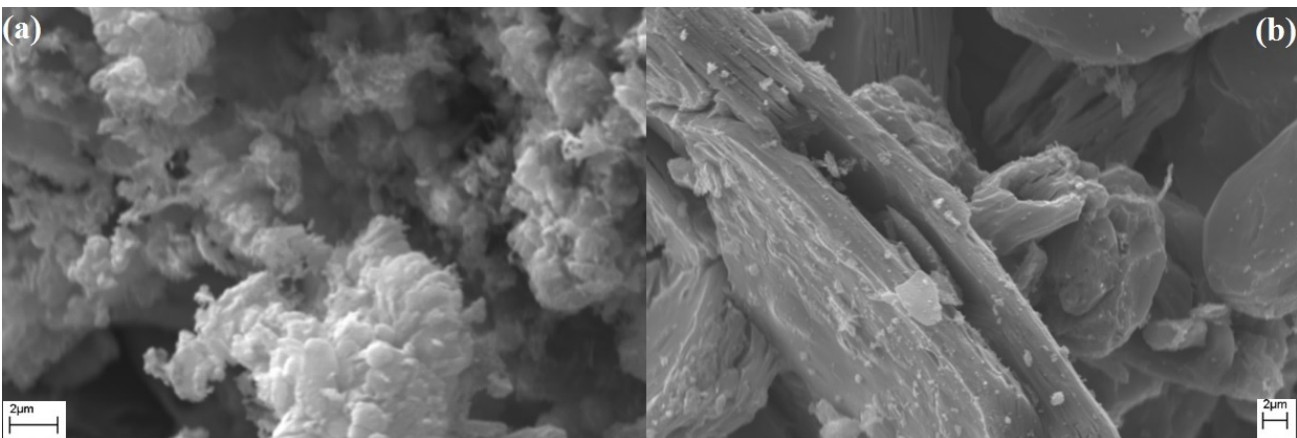

**Figure 13.** SEM images of the decomposition products form from compound **1** at 500 °C under nitrogen (**a**) and air (**b**).

**Table 3.** The specific surface area ($m^2 \cdot g^{-1}$) of the thermal decomposition intermediates and end-products that formed at different temperatures from compound **1** in $N_2$ and air.

| Temperature, °C | | 115 | 135 | 160 | 250 | 390 | 500 |
|---|---|---|---|---|---|---|---|
| Specific surface area, $m^2 \cdot g^{-1}$ | Under $N_2$ | 6 | 17 | 16 | 7 | 27 | 8 |
| | Under air | – | 6 | 4 | 4 | – | 35 |

The specific surface area (SSA) of the thermal decomposition intermediates and products that formed from compound **1** under $N_2$ changed irregularly with increasing sintering temperature. The reaction product prepared at 390 °C had the highest surface area (~27 $m^2 \cdot g^{-1}$). Increasing the temperature to 500 °C led to a decrease in the surface area (8 $m^2 \cdot g^{-1}$) due to sintering during crystallization. In the presence of air, the highest SSA value was found at 500 °C (35 $m^2 \cdot g^{-1}$), which shows less compactness of the spinel lattice in air than in $N_2$ due to the oxidation of the residual reducing components and possible gas formation during the crystallization process.

We checked the photocatalytic potential of the thermal decomposition intermediates and products of compound **1** in the degradation of two organic dyes below and above pH values relates to their $pK_a$ values. The photocatalytic activity results are summarized in Figure 14, and the apparent decomposition rate ($k_{app}$, pseudo-first order rate constant) for each intermediate can be seen in Table 4 and Figure 14.

The studied catalysts that formed under an inert atmosphere with the thermal decomposition of compound **1** showed good activity in accelerating (11–54 times) the degradation of Congo red at pH = 5.7 (alkaline form of Congo red). The highest value was found for the intermediate that formed at 115 °C (probably a todorokite-like material) in the first decomposition step under $N_2$. Using the similar todorokite-like material that formed from compound **2** at 125 °C resulted in less catalytic activity (5 and 18 times, under $N_2$ and air, respectively) for the photodegradation of Congo red under analogous conditions [9].

Increasing the decomposition temperature of compound **1** decreases the activity of the products. Catalytic activity does not correlate with the specific surface area; thus, the changes in catalytic activity may be attributed to the changes of the valence and defect distributions in the intermediates formed at various temperatures. The decomposition products that formed at 500 °C (cubic $Co_{1.5}Mn_{1.5}O_4$) in air or $N_2$ gave the same results: nine times faster decomposition. Neither the specific surface area nor the atmosphere of the synthesis had any influence on the catalytic activity of the $Co_{1.5}MnO_{1.5}O_4$ spinel that formed at 500 °C. It is different from the result found in the case of tetragonal $CoMn_2O_4$ spinel that formed from compound **2**. Here, the atmosphere influenced the photocatalytic

acidity in Congo red degradation (13 and 9 times, samples prepared under $N_2$ and air, respectively) [9].

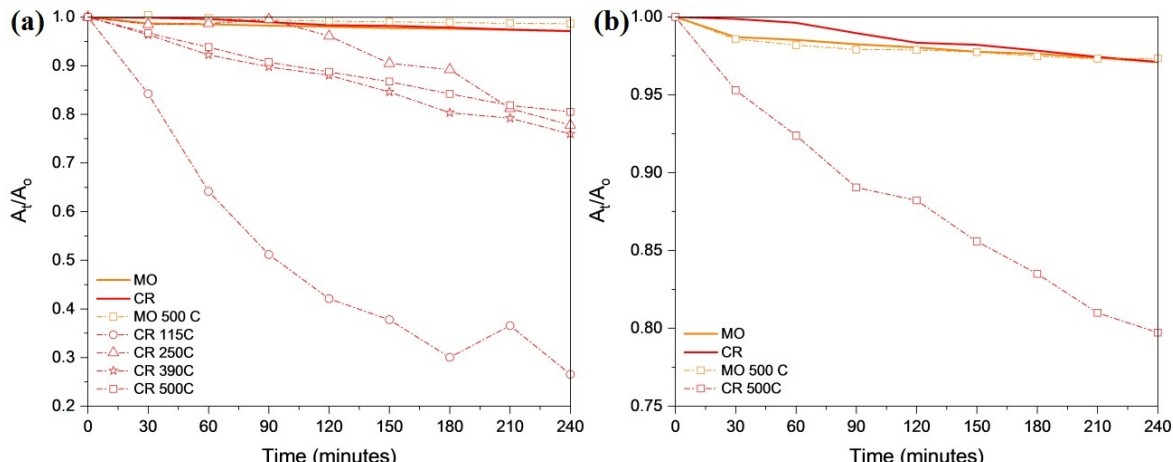

**Figure 14.** Photocatalysis of decomposition intermediates and products forming from compound **1** at various temperatures under nitrogen (**a**) and air (**b**) during the photodecomposition of Congo red. Figure 14 shows that among the two studied dyes, Methyl orange and Congo red; only the degradation of Congo red was catalyzed by these catalyst candidates. The pH of the dye solutions was adjusted to be over the $pK_a$ values of dyes.

**Table 4.** Photocatalytic parameters ($k_{app}$) in the decomposition of Congo red and Methyl orange catalyzed with intermediates forming under $N_2$ and end-products forming under air or $N_2$ in different conditions.

| Dyes ($2 \cdot 10^{-5}$ M) and Catalysts Parameters | pH | $K_{app}/10^{-4}$, $min^{-1}$ | $R^2$ |
|---|---|---|---|
| Congo red, without catalyst | 5.7 | 1.0 | 0.98 |
| Congo red, catalyst made at 115 °C in $N_2$ | 5.7 | 54.0 | 0.93 |
| Congo red, catalyst made at 250 °C in $N_2$ | 5.7 | 12.0 | 0.87 |
| Congo red, catalyst made at 390 °C in $N_2$ | 5.7 | 11.0 | 0.99 |
| Congo red, catalyst made at 500 °C in $N_2$ | 5.7 | 9.0 | 0.99 |
| Congo red, catalyst made at 500 °C in air | 5.7 | 9.0 | 0.98 |
| Methyl orange, without catalyst | 5.6 | 0.8 | 0.99 |
| Methyl orange, catalyst made at 500 °C in $N_2$ | 5.6 | 0.6 | 0.97 |
| Methyl orange, catalyst made at 500 °C in air | 5.6 | 0.6 | 0.94 |

## 3. Materials and Methods

Chemical-grade $CoCl_2.6H_2O$, ammonium chloride, hexachloroplatinic acid ($H_2PtCl_6.6H_2O$), sodium permanganate (40% aq. soln.), 25% aq. ammonia, 37% aq. hydrochloric acid, silver nitrate, and NaOH were supplied by Deuton-X Ltd. (Érd, Hungary).

### 3.1. Synthesis of Compound **1**

Compound **1** was prepared following Klobb's method [31] by a direct combination of 1 g of compound $[Co(NH_4)_3](MnO_4)_3$ and 4.13 g of $[Co(NH_3)_6]Cl_3$ in 70 mL of water (60.0 °C). The mixture was stirred for 15 min and left to be chilled in a fridge, where blocks of small strips of crystals formed at around 1.0 °C. Then, the formed crystals were filtered off and dried in a desiccator at room temperature.

### 3.2. Preparation of Co(NH₃)₆Cl₃ (Compound **4**)

[Co(NH₃)₆]Cl₃ was prepared according to the method of Jörgensen [73]. First, 14 g of NH4Cl was dissolved in 20 mL of distilled water; then, 21.6 g of CoCl₂·6H₂O was added to the solution and stirred for fifteen minutes. After that, 0.5 g of activated carbon (powder) was added with 55 mL of cc. ammonia solution. An O₂ stream was passed into the solution slowly. In every 15 min interval, 2–3 mL of cc. ammonia was added. The reaction can be considered completed when the color completely turns to brown. The precipitate was filtered out, 100 mL of hydrochloric acid (~5%) was added, and the mixture was heated to 80.0 °C and kept at that temperature for fifteen minutes. Then, the mixture was filtered while still hot, and 29 mL of cc. HCl was added, after which [Co(NH₃)₆]Cl₃ crystallized out on chilling the solution to 0.0–5.0 °C. The crystals were washed with distilled water and dried at room temperature.

### 3.3. Preparation of [Co(NH₃)₆](MnO₄)₃ (Compound **3**)

Compound **3** was prepared by dissolving 1 g of [Co(NH₃)₆]Cl₃ in 50 mL of distilled water, then by adding of 40% aq. NaMnO₄ (2.29 mL). The mixture was heated at 60.0 °C for 15 min. The mixture was left to cool down to form compound **3**. The crystals were filtered off and washed with cold distilled water.

### 3.4. Analytical Methods

The classical analytical and basic instrumental measurements were taken using methods and instruments described in detail in our earlier papers [1–7]. The most essential conditions of the measurement methods are listed below.

### 3.5. Elemental Analysis

The Co and Mn content of compound **1** was determined by ICP−OES (atomic emission spectroscopy with a Spectro Genesis ICP−OES instrument (SPECTRO Analytical Instruments GmbH, Kleve, Germany). A multielement standard (Merck Chemicals GmbH, Darmstadt, Germany) was used for calibration. Chloride content was determined by argentometric titration. The ammonia and ammonium ion content of the decomposition intermediates were determined by gravimetry in the form of (NH₄)₂PtCl₆. Gaseous ammonia was liberated from ammonium salts by a 10% aqueous solution of NaOH with subsequent boiling. The classic method to determine NH₃ via absorption in a HCl solution using back-titration does not work here because of the NOₓ as HNO₃ precursor formation during the decomposition of compound **1**. The NOₓ gases do not give a precipitate with H₂PtCl₆.

### 3.6. Vibrational Spectroscopy

The far−IR and mid−IR spectra of compound **1** were recorded in attenuated total reflection (ATR) mode using a BioRad−Digilab FTS−30−FIR and a Bruker Alpha IR spectrometer for the 400–40 and 4000–400 cm$^{-1}$ ranges, respectively. Raman spectroscopy at 298 and 123 K between 2000 and 200 cm$^{-1}$ was performed on a Horiba Jobin−Yvon LabRAM microspectrometer. Two external laser sources were used (a 785 nm diode laser and a 532 nm Nd:YAG laser with ~80 and ~40 mW, respectively), and an Olympus BX−40 optical microscope. A Linkam THMS600 temperature-controlled microscope stage was used in the low-temperature measurements. The laser beam was focused on an objective of 20×. Due to the heat sensitivity of compound **1,** a D0.6 (123 K) and a D2 (298 K) intensity filter was used with a 785 nm laser to decrease the laser power to 25% and 1%, respectively. The 532 nm excitation required a D3 (298 K) density filter using only 0.1% of the initial light power. A confocal hole of 1000 μm and a monochromator with 950 (for the diode laser) and 1800 groove mm$^{-1}$ (for the Nd:YAG laser) gratings were used to disperse light. The resolution was 4 cm$^{-1}$, and the exposure times were 20–200 s.

### 3.7. UV−Vis Spectroscopy

The room temperature UV−VIS diffuse reflectance spectrum of compound **1** was measured with a Jasco V−670 UV–VIS instrument equipped with a NV−470 integrating sphere ($BaSO_4$ was used as standard).

### 3.8. Scanning Electron Microscopy

Scanning electron microscopy (SEM) was performed with a JEOL JSM−5500LV instrument. The samples were fixed on a Cu/Zn alloy with a carbon tape holder and sputtered with a conductive Au/Pd layer for imaging.

### 3.9. Powder X-ray Diffractometry

Powder X-ray tests were performed with a Philips PW−1050 Bragg–Brentano parafocusing goniometer equipped with a copper cathode (40 kV, 35 mA, secondary beam graphite monochromator, proportional counter). Scans were recorded in step mode, and the diffraction patterns were evaluated with a full profile fitting technique.

### 3.10. Single-Crystal X-ray Diffraction

A clear red platelet-like crystal of $[Co(NH_3)_6]Cl_2MnO_4$ was mounted on a loop. Cell parameters were determined by least squares using 22,083 ($3.160 \leq \theta \leq 27.550$) reflections. Intensity data were collected on a Rigaku R−Axis Rapid diffractometer (monochromator; Mo−$K_\alpha$ radiation, $\lambda = 0.71073$ Å) at 163 (2) K in the range $3.157 \leq \theta \leq 27.473$. A total of 18,320 reflections were collected of which 2684 were unique [$R$(int) = 0.0588, $R(\sigma)$ = 0.0372]; intensities of 2351 reflections were greater than $2\sigma(I)$. Completeness was measured to $\theta = 0.997$.

A numerical absorption correction was applied to the data (the minimum and maximum transmission factors were 0.868 and 0.987).

The structure was solved by iterative methods (and subsequent difference syntheses).

The anisotropic full-matrix least-squares refinement on $F^2$ for all non-hydrogen atoms yielded $R_1 = 0.0397$ and $wR_2 = 0.0747$ for 1332 [$I > 2\sigma(I)$] and $R_1 = 0.0422$ and $wR_2 = 0.0854$ for 1332 [$I > 2\sigma(I)$] and $R_1 = 0.0528$ and $wR_2 = 0.0892$ for all (2684) intensity data, (number of parameters = 136, goodness-of-fit = 1.152, the maximum and mean shift/esd are 0.000 and 0.000, respectively). The maximum and minimum residual electron density in the final difference map were 1.535 and $-0.574$ e.Å$^{-3}$, respectively. The weighting scheme applied was $w = 1/[\sigma^2(F_o^2) + (0.03461.8733P)^2 + 1.8733P]$ where $P = (F_o^2 + 2F_c^2)/3$.

Hydrogen atomic positions were calculated from assumed geometries. Hydrogen atoms were included in structure factor calculation, but they were not refined. The isotropic displacement parameters of the hydrogen atoms were approximated from the $U$(eq) value of the atom they were bonded to.

The CSD Deposition Number is 2220607.

### 3.11. Thermal Studies

The TG/MS measurements were performed with a modified TGS−2 thermobalance (Perkin Elmer, Waltham, MA, USA) coupled to a HiQuad quadrupole mass spectrometer (Pfeiffer Vacuum, Germany). A ~1 mg sample was measured in a Pt sample pan. The decomposition process was followed from 25 to 500 °C with a 2 °C min$^{-1}$ heating rate, in Ar carrier gas at a flow rate of 140 cm$^3$ min$^{-1}$. Selected ions were selected for monitoring (SIM) in a range of $m/z$ = 2–88.

The DTG data were collected by a TA Instruments SDT Q600 thermal analyzer. A 2 mg sample was heated from 25 to 150 °C with a heating rate of 2 °C/min and from 150 to 500 °C at 5 °C min$^{-1}$. The gas flow of the inert (nitrogen) and oxidative (synthetic air) gases was 20 mL/min.

The DSC curves were recorded between $-130$ and 300 °C with a Perkin Elmer DSC 7 instrument with a sample mass of 3–5 mg and a heating rate of 5 °C/min under a continuous nitrogen or oxygen flow (20 cm$^3$ min$^{-1}$) in an unsealed aluminum pan.

### 3.12. Measurement of Magnetic Susceptibility

Magnetic measurements were carried out with an MSB$-$MKI magnetic susceptibility balance (Sherwood Scientific Ltd., Cambridge, UK) calibrated with $Hg[Co(NCS)_4]$ standard.

### 3.13. Photocatalytic Measurements

To evaluate the photocatalytic activity of the samples, 1.0 mg of the decomposition products from compounds **1** and **2** were put into 3 mL of an aqueous solution of Methyl Orange (MO—$4 \times 10^{-5}$ M), and Congo Red (CR—$2 \times 10^{-5}$ M) poured into quartz cuvettes. The samples were kept in the dark overnight for the adsorption equilibrium. After that, they were submitted to a UV irradiation provided by Osram 18 W blacklight lamps ($\lambda$ = maximum intensity at 375 nm). The cuvettes were placed 5 cm from each lamp, and the absorbance was measured every 30 min during four hours by a Jasco V$-$550 UV$-$VIS spectroscope. The relative absorbance values of the most intensive peaks for MO (464 nm) and CR (497 nm), below and above their $pK_a$ value, were considered to evaluate the catalysts' activity in the degradation of dyes. The dilute perchloric acid solution (0.1 M) was used to decrease the pH of both dyes.

### 4. Conclusions

We synthesized [hexaamminecobalt(III)] dichloride permanganate $[Co(NH_3)_6]Cl_2(MnO_4)$ (compound **1**) in the reaction of $[Co(NH_3)_6]Cl_3$ and $[Co(NH_3)_6](MnO_4)_3$. Compound **1** was spectroscopically (FT$-$IR, far$-$IR, Raman and UV) characterized. The structure of compound **1** was determined by single-crystal X-ray diffraction. The 3D hydrogen bond network (N–H$\cdots$O–Mn and N–H$\cdots$Cl interactions) are centers of a solid-phase redox reaction that occurs between the permanganate anion and ammonia ligand. The $Co^{III}$ centers and permanganate ions act as oxidant, and chloride ions and ammonia act as reducing agents in consecutive redox reactions during heating. The temperature-limited thermal decomposition of compound **1** under boiling toluene resulted in the formation of $(NH_4)_4Co_2Mn_6O_{12}$ (a todorokite-like manganese oxide network ($Mn^{II}_4Mn^{III}_2O_{12}{}^{10-}$) with square-shape channels). Aqueous leaching of the residue gave $[Co(NH_3)_6]Cl_3$, $[Co(NH_3)_6]Cl_2$, $NH_4NO_3$ and $NH_4Cl$ as water-soluble products.

The heat treatment products of the solid **1** and **2** at 500 °C are cubic and tetragonal spinels with $Co_{1.5}Mn_{1.5}O_4$ ($MnCo_2O_4$ type) and $CoMn_2O_4$ composition, respectively. Heating of the residues of thermal decomposition of compounds **1** and **2** under toluene at 110 °C followed with aqueous leaching and heating at 500 °C for 1.5 h gave the same tetragonal spinel phase with the $Co_{0.75}Mn_{2.25}O_4$ formula. Without aqueous leaching, the products of heating at 500 °C are tetragonal $Co_{1.5}Mn_{1.5}O_4$ and $CoMn_2O_4$, respectively.

The cubic $Co_{1.5}Mn_{1.5}O_4$ prepared from compound **1** at 500 °C catalyzes the degradation of Congo red with UV light. The $((NH_4)_4Co_2Mn_6O_{12})$ intermediate prepared from compound **1** under $N_2$ degraded Congo red 54 times faster, which is much faster than what was found for $((NH_4)_4Co_2Mn_6O_{12})$ that formed from compound **2** under $N_2$.

**Supplementary Materials:** The following supporting information can be downloaded at: https://www.mdpi.com/article/10.3390/inorganics10120252/s1, Figure S1: The powder X-ray diffractogram of compound **1**; Figure S2: The calculated (from SXRD data) powder X-ray diffractogram of compound **1**; Table S1: Crystal data and structure refinement of compound **1**; Table S2: The hydrogen bond interactions in the crystal structure of hexaamminecobalt(III) dichloro permanganate; Table S3: The bond lengths (Å) and angles (°) in the crystal structure of hexaamminecobalt(III) dichloro permanganate; Table S4: The IR and Raman spectral data of permanganate ion in compound **1**; Table S5: The IR and Raman spectral data of the ammonia ligand in compound **1**; Table S6: The IR and Raman spectral data of the CoN6 skeleton in compound **1**; Table S7: Electronic transitions (in nm) of the hexaamminecobalt(III) cation in compound **1** and in octahedral and trigonally distorted (compressed) octahedral structures; Figure S3: The group analysis for Co atoms in compound **1**; Figure S4: The group analysis for Cl atoms in compound **1**; Figure S5: The IR spectra of compound **1**; Figure S6: The IR spectra of compound **1** between 950 and 600 cm$^{-1}$; Figure S7: The far-range IR spectra of compound **1**; Figure S8: The Raman spectra (at room temperature) of compound **1** with (a) 532 and (b) 785 nm excitation; Figure S9: The Raman spectra (at 123K) of compound 1 with (a) 532

and (b) 785 nm excitation; Figure S10: The (a) full range and (b) 200–570 nm range UV−VIS spectra of compound 1; Figure S11: The IR spectra of the evaporation residue from the aqueous extract of the decomposition intermediate made in boiling toluene from compound **1**; Figure S12: The powder XRD of the evaporation residue from the aqueous extract of the decomposition intermediate made in boiling toluene from compound **1**; Figure S13: The IR spectra of the water-insoluble decomposition residue of the decomposition intermediate made in boiling toluene from compound **1**; Figure S14: The powder X-ray diffractograms of the decomposition intermediates and products of compound **2**: (a) made by heat-treatment in toluene at 110 °C, (b) made by the heat treatment of compound **1** at 500 °C in air and (c) made from the water-insoluble part of the decomposition intermediate of compound **1** made in toluene by heating at 500 °C and (d) made from the decomposition intermediate of compound **1** made in toluene by heating at 500 °C.

**Author Contributions:** Conceptualization, L.K.; formal analysis, V.M.P. and L.B.; investigation, L.B., K.A.B., A.F., F.P.F., Z.C., I.M.S. and B.B.H.; writing—original draft preparation, L.K.; writing—review and editing, L.B., V.M.P. and K.A.B., visualization, L.B. and K.A.B., supervision, L.K. All authors have read and agreed to the published version of the manuscript.

**Funding:** The research was supported by the European Union and the State of Hungary, co-financed by the European Regional Development Fund (VEKOP-2.3.2-16-2017-00013) (L.K.) and the ÚNKP-21-3 and 22-3 New National Excellence Program of the Ministry for Innovation and Technology from the source of the National Research, Development and Innovation Fund (K.A.B.) and the National Research Development and Innovation Office through OTKA grant K124544. B.B.H. thanks to the Ministry of Education, Science, and Technological Development of the Republic of Serbia (Grant No. 451-03-68/2022-14/200125) for funding.

**Institutional Review Board Statement:** Not applicable.

**Informed Consent Statement:** Not applicable.

**Data Availability Statement:** Not applicable.

**Conflicts of Interest:** The authors declare no conflict of interest.

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
