# Peer review of "[Hexaamminecobalt(III)] Dichloride Permanganate—Structural Features and Heat-Induced Transformations into (CoII,MnII)(CoIII,MnIII)2O4 Spinels"

_inorganics, doi:10.3390/inorganics10120252_

Round 1
Reviewer 1 Report
The paper of László Kótai and co-authors is a fundamental work on obtaining known hexaamminecobalt(III) dichloride permanganate [Co(NH3)6]Cl2(MnO4) and investigating it with a great deal of known physicochemical research methods. The authors isolated compound 1 in single crystal form, the crystal structure was determined. The manuscript shows how much new information can be obtained from previously synthesized and known compound. The authors showed that cobalt complex 1 can be used for the synthesis of heterometallic spinel-like structures.
The list of references consists of 61 papers, 22 of which are self-citations (at least a complete match by name, I did not check in Scopus). The MDPI publishing house does not restrict self-citation. But I advise to rewrite the introduction, using references to the work of other authors. Because
“The preparation and thermal decomposition of transition metal complexes with reducing ligands and oxygen-containing anions are intensively studied areas of coordination chemistry (1-12)” all 12 references are self-citation, this does not prove that the field of thermal decomposition of coordination compounds is indeed being studied by many scientific groups.
In some places the article seems to be over investigated, much information can be removed in supplementary materials to reduce total amount pages of the article for better perception, e.g. 2.3 Spectroscopic Properties of Compound 1.
Powder X-ray diffractograms (Figure 11, a and b) have low quality with noise. I advise to remove a and b to ESI or exclude the information.
Page 17
6[Co(NH3)6]Cl2MnO4 = 4MnIICoIII1.5MnIII1.5O4 + 2N2O + 6H2O+ 28NH3 + 12HCl +2N2
3Co(NH3)5Cl](MnO4)2 = 3CoMn2O4 + N2O + 9H2O+ 8NH3 + 3HCl +2.5 N2
To prove the the reaction, gas chromatography can be done.
Page 21, Materials and Methods – why did you need silver nitrate?
Typos:
Page 6 indicated by red in the fingerprint plots) – round bracket is absent
Page 10 [Co(NH3)6](3+) – 3+ should be upper case (twice, next sentence)
Page 11 products containing N are oxidized – should be N2 or nitrogen atom
Page 14 Generally, cubic spinels with inverse spinel structures generally form when more – generally-generally
Page 19 the decomposition products form from – remove form?
Page 21 hot and 40 29 mL - ?
In general, the article is well written, there is enough data for publication in Inorganics (one structure, one complex). The current work seems interesting to me and I recommend it to publish in Inorganics in a section “Coordination Chemistry”.
Author Response
First of all, we would like to express our sincere thanks to the reviewer for improving our manuscript. All of the suggestions were accepted and the text was modified accordingly.
The list of references consists of 61 papers, 22 of which are self-citations (at least a complete match by name, I did not check in Scopus). The MDPI publishing house does not restrict self-citation. But I advise to rewrite the Introduction, using references to the work of other authors. Because “The preparation and thermal decomposition of transition metal complexes with reducing ligands and oxygen-containing anions are intensively studied areas of coordination chemistry (1-12)” all 12 references are self-citation, this does not prove that the field of thermal decomposition of coordination compounds is indeed being studied by many scientific groups.
Twelve new references are added to show that other research groups are interested in this field. I have to make a remark that the self-citations are used in other parts of the manuscript (e.g., giving information about a comparable compounds or method). Since generally more than 10 laboratories cooperate with each other to make our papers, combining their results (spectroscopic, thermal, X-ray, magnetic, etc.) to characterize the compounds completely, it is easy to understand the fact that one or more authors among these labs will be the author in another paper of the field. We do not want to fragment the knowledge about a compound into 3-4 smaller papers with fewer authors (in this case, a lot of these citations would be independent) because the fragmentation would cause worse visibility of the relations between the properties of the materials. Also, fragmentation of the work would be unethical.
In some places, the article seems to be over investigated, much information can be removed in supplementary materials to reduce total amount pages of the article for better perception, e.g., 2.3 Spectroscopic Properties of Compound 1.
A part of the mentioned chapter (three Tables) is shifted to ESI. But, I feel the part left back is important to be kept in the paper to show and interpret the spectroscopic properties.
Powder X-ray diffractograms (Figure 11, a and b) have low quality with noise. I advise to remove a and b to ESI or exclude the information.
Yes, this is only noise in Fig. 11a because the sample is an amorphous material. Removing the diffractograms a and b would result in losing the information about the lack of peaks(a)/improving the peak/noise relation (b,c). We want to show that on heating, the completely amorphous materials are transformed (with appearance of peaks) and finally turn into crystalline material (d) with a good peak/noise ratio. It is this reason why we feel it should be left in the paper.
Page 17
6[Co(NH3)6]Cl2MnO4 = 4MnIICoIII1.5MnIII1.5O4 + 2N2O + 6H2O+ 28NH3 + 12HCl +2N2
3Co(NH3)5Cl](MnO4)2 = 3CoMn2O4 + N2O + 9H2O+ 8NH3 + 3HCl +2.5 N2
To prove the the reaction, gas chromatography can be done.
The TG-MS measurements were done with an instrument that has an inner chromatographic column, and these components were separated on this GC column and detected by MS-based at various m/z values.
Page 21, Materials and Methods – why did you need silver nitrate?
The silver nitrate was used in the argentometric determination of the chloride content to check the purity of compound 1. It has been inserted into the experimental part.
Typos:
Page 6 indicated by red in the fingerprint plots) – round bracket is absent
It has been revised.
Page 10 [Co(NH3)6](3+) – 3+ should be upper case (twice, next sentence)
It has been revised.
Page 11 products containing N are oxidized – should be N2 or nitrogen atom
It has been revised.
Page 14 Generally, cubic spinels with inverse spinel structures generally form when more – generally-generally
It has been revised.
Page 19 the decomposition products form from – remove form?
It has been revised.
Page 21 hot and 40 29 mL - ?
It has been revised.
Reviewer 2 Report
This manuscript reports on the structural features and heat-induced transformations of [Co(NH3)6]Cl2(MnO4) compound into (CoII,MnII)(CoIII,MnIII)2O4 spinels. The manuscript is well-organized. The article can help to comprehend the transformation processes of precursors during the synthesis of materials. The main problem with the manuscript is the insufficient number of references to other research works demonstrating the state-of-the-art.
The reviewer has some comments and recommendations:
-
The authors could shorten the title '[Hexaamminecobalt(III)] dichloride permanganate – Structural Features and Heat-induced Transformations into (CoII,MnII)(CoIII,MnIII)2O4 Spinels'
-
The Abstract is excessively detailed. The Abstract should reflect the main problem that the authors tried to solve, their motivation and purpose of the research, the short results and findings. Please, rewrite the abstract.
-
In Introduction, the reviewer recommends adding the recent references of other research groups, not only authors' articles, to show this study's relevance in the field of coordination chemistry. The references [1-15] are articles of the manuscript's authors.
-
For ATR-IR spectroscopy, the reflection or absorbance is usually specified on the y-axis. The absorbance should be in Fig.10 and ESI 6,7.
-
The manuscript has a high percentage of self-citation. 22 of the 61 articles of manuscript's authors are given in References.
-
'Conclusions' paragraph is a rewritten version of the Abstract. Conclusions should include a generalized list of the main results of the study as well as an assessment of their significance for science. It is not necessary to mention the research methods again and describe in detail the results. They should be revised.
Finally, after minor corrections, the manuscript may be accepted for publication in Inorganics.
Author Response
First of all, we would like to express our sincere thanks to the reviewer for improving our manuscript. All of the suggestions were accepted and the text was modified accordingly.
This manuscript reports on the structural features and heat-induced transformations of [Co(NH3)6]Cl2(MnO4) compound into (CoII,MnII)(CoIII,MnIII)2O4 spinels. The manuscript is well-organized. The article can help to comprehend the transformation processes of precursors during the synthesis of materials. The main problem with the manuscript is the insufficient number of references to other research works demonstrating the state-of-the-art.
We inserted 12 publications as recently published new references from other laboratories into the Introduction
The reviewer has some comments and recommendations:
- The authors could shorten the title '[Hexaamminecobalt(III)] dichloride permanganate – Structural Features and Heat-induced Transformations into (CoII,MnII)(CoIII,MnIII)2O4 Spinels'
It has been done.
- The Abstract is excessively detailed. The AbstractAbstract should reflect the main problem that the authors tried to solve, their motivation and purpose of the research, the short results and findings. Please, rewrite the abstract.
It has been done.
- In Introduction, the reviewer recommends adding the recent references of other research groups, not only authors' articles, to show this study's relevance in the field of coordination chemistry. The references [1-15] are articles of the manuscript's authors.
Ten new references have been added. I have to remark that the citations 1-15 are also used in other parts of the manuscript paper (e.g. giving information about a comparable compound or method). Since more than 10 laboratories cooperate with each other to make 1-1 of our papers, combining their results (spectroscopic, thermal, X-ray, magnetic, etc.), it is easy to face the fact that one or more authors of these labs will be an author in another paper of the field.
- For ATR-IR spectroscopy, the reflection or absorbance is usually specified on the y-axis. The absorbance should be in Fig.10 and ESI 6,7.
It has been done.
- The manuscript has a high percentage of self-citation. 22 of the 61 articles of manuscript's authors are given in References.
The number of references changed to 73, and the self-citations were used to give some comparison, method, or data about a similar compound synthesized by us. E.g., we did not want to repeat the experimental methods in detail. Therefore, we cited our previous work to give information on how we did the measurement, or compared the new results of this compound with the data measured by us about other similar compounds.
- 'Conclusions' paragraph is a rewritten version of the Abstract. Conclusions should include a generalized list of the main results of the study as well as an assessment of their significance for science. It is not necessary to mention the research methods again and describe in detail the results. They should be revised.
It has been revised.
Reviewer 3 Report
Hexa-ammine cobalt complexes as permanganate adducts have been studied experimentally. The manuscript describes synthesis, as well as crystallographic and spectroscopic characterization of the material. Special interest has been on the thermal decomposition of the synthesized complex, and the redox properties of the resulting compounds. The synthetic procedure of compound 1 is already known, but the detailed characterization is missing, and the present paper provides such analysis. The study is very thorough, and the manuscript is clearly and consistently written, although it is a bit long and would benefit from slight condensing. I have only one minor comment, which the authors should address before accepting for publication:
In the abstract, the authors define compound 1, but not compound 2, and yet they speak about the properties of both. This can be confusing. In addition, it is not really obvious, where does compound 2 come from, since it has not described in detail. Was it synthesized separately, or was it a side product? Or is it something you can buy in a bottle? Please clarify.
Author Response
First of all, we would like to express our sincere thanks to the reviewer for improving our manuscript. All of the suggestions were accepted and the text was modified accordingly.
In the abstract, the authors define compound 1, but not compound 2, and yet they speak about the properties of both. This can be confusing. In addition, it is not really obvious, where does compound 2 come from, since it has not described in detail. Was it synthesized separately, or was it a side product? Or is it something you can buy in a bottle? Please clarify.
The Abstract has been changed. Compound 2 was synthesized separately by us as described in one of our previous papers (citation is given).
Reviewer 4 Report
The article is devoted to obtaining heterometallic precursors for the synthesis of bimetallic oxide materials. The work is performed at a decent level using various methods of analysis and is well structured. I would recommend some changes to the manuscript to improve the presentation of the work itself.
First, the abstract should be cut at least in half, leaving only the most important information.
Secondly, when describing XRD complexes, it is necessary to present distances between metal centers, e.g., between Mn-Mn, Co-Co, Mn-Co. At the same time, it is worth specifying the smallest possible distance. Third, it is worth using the SHAPE program to analyze the geometry of the metallocenter polyhedra.
Fourth, compound 1 can cause the oxidation of toluene through the methyl group due to the presence of permanganate anion. Was the organic phase analyzed by heating product 1 in toluene?
In the introduction, I would focus on obtaining heterometallic oxides based on bi- or trimetallic molecular precursors. Consider including references such as: DOI: 10.3390/molecules27227894; DOI: 10.1039/C2CS35296F.
I did not see magnetic measurement data and explanations for heterometallic compounds in the article. What information magnetism provides to establish the structure remains unclear.
The section on oxidation of dyes with heterometallic oxide is described too briefly. What concentrations of catalyst and dye were used? It is best to recalculate in moles rather than limiting yourself to values for solutions.
Author Response
First of all, we would like to express our sincere thanks to the reviewer for improving our manuscript. All of the suggestions were accepted and the text was modified accordingly.
First, the Abstract should be cut at least in half, leaving only the most important information.
The Abstract has been shortened.
Secondly, when describing XRD complexes, it is necessary to present distances between metal centers, e.g., between Mn-Mn, Co-Co, Mn-Co. At the same time, it is worth specifying the smallest possible distance.
No direct metal-metal interactions were found, the shortest Co-Co distance is 7.198(1) Å, the shortest Co-Mn distance is 5.011(1) Å, and the shortest Mn-Mn distance is 6.895(1) Å in the structure. A sentence is inserted about it into the text.
Third, it is worth using the SHAPE program to analyze the geometry of the metallocenter polyhedra.
A picture of the metallocenter polyhedra is inserted. We could make it by Mercury, we have no SHAPE software.
Fourth, compound 1 can cause the oxidation of toluene through the methyl group due to the presence of permanganate anion. Was the organic phase analyzed by heating product 1 in toluene?
Yes, we checked the oxidation of toluene with compounds 1 and 2, even before using that as solvent to study the decomposition of these compounds. We used the permanganate compounds 1 and 2 in the solid state, and as solids they are weak oxidants and do not oxidize toluene at all, even upon heating. In their aq. solution, especially at acidic pH, they would oxidize toluene into benzoic acid, of course, because that is the reaction of the free (dissolved) permanganate ion.
Similar results were found for several other solid complex permanganates prepared by us. They were weak oxidants, could oxidize only the strongly reducing organic substrates as benzyl alcohols or sulfides, into benzaldehyde or disulfides, respectively, but not the aromatic or chlorinated solvents used as media for these reactions. The relative "inertness" of the solid complex permanganates toward hydrocarbon and chlorinated solvents is typical enough.
In the Introduction, I would focus on obtaining heterometallic oxides based on bi- or trimetallic molecular precursors. Consider including references such as: DOI: 10.3390/molecules27227894; DOI: 10.1039/C2CS35296F.
These have been inserted. The Introduction has been modified.
I did not see magnetic measurement data and explanations for heterometallic compounds in the article. What information magnetism provides to establish the structure remains unclear.
We have done room temperature magnetic susceptibility measurements on the oxides, but the defect structures (changes in molar masses) and other reasons led us to the decision, that – as is written at the end of the manuscript – the detailed characterization of these heterobimetallic materials together with their catalytic activity in some industrially important reactions will be published in a separated forthcoming paper.
The section on the oxidation of dyes with heterometallic oxide is described too briefly. What concentrations of catalyst and dye were used? It is best to recalculate in moles rather than limiting yourself to values for solutions.
The reviewer is right. We forgot to insert the photocatalytic measurement's conditions. It has been done; a new paragraph was added to the experimental section. The results are given in the usual forms to compare the photocatalytic activities with the results given in other papers.
Round 2
Reviewer 4 Report
The authors have fully responded to the reviewers' comments and I am pleased to be able to recommend this paper for publication in Inorganics.